# SAME: Stabilized Mixture-of-Experts for Multimodal Continual Instruction Tuning

**Zhen-Hao Xie** [* 1 2]  **Jun-Tao Tang** [* 2]  **Yu-Cheng Shi** [2]  **Han-Jia Ye** [1 2]  **De-Chuan Zhan** [1 2]  **Da-Wei Zhou** [1 2]

## Abstract

Multimodal Large Language Models (MLLMs) achieve strong performance through instruction tuning, but real-world deployment requires them to continually expand their capabilities, making Multimodal Continual Instruction Tuning (MCIT) essential. Recent methods leverage sparse expert routing to promote task specialization, but we find that the expert routing process suffers from drift as the data distribution evolves. For example, a grounding query that previously activated localization experts may instead be routed to irrelevant experts after learning OCR tasks. Meanwhile, the grounding-related experts can be overwritten by new tasks and lose their original functionality. Such failure reflects two problems: *router drift*, where expert selection becomes inconsistent over time, and *expert drift*, where shared experts are overwritten across tasks. Therefore, we propose StAbilized Mixture-of-Experts (SAME) for MCIT. To address router drift, SAME stabilizes expert selection by decomposing routing dynamics into orthogonal subspaces and updating only task-relevant directions. To mitigate expert drift, we regulate expert updates via curvature-aware scaling using historical input covariance in a rehearsal-free manner. SAME also introduces adaptive expert activation to freeze selected experts during training, reducing redundant computation and cross-task interference. We also introduce a new benchmark to evaluate MCIT with long task sequences, and extensive experiments demonstrate SAME's SOTA performance. Code is available at https://github.com/LAMDA-CL/Prism.

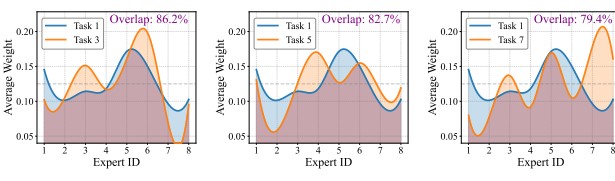

*(a)* Task 1 vs Task 3  *(b)* Task 1 vs Task 5  *(c)* Task 1 vs Task 7

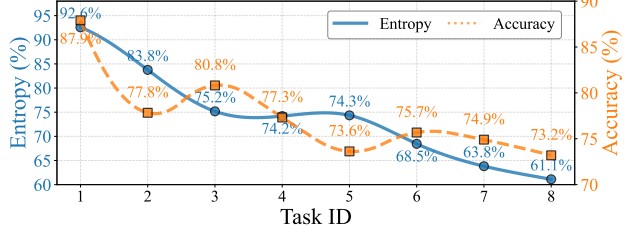

*(d)* Dynamics of entropy and accuracy for re-routing.

*Figure 1.* (a~c) On the Task 1 test set, the router's expert-activation distribution shifts as new tasks are learned, with decreasing overlap against later-task routers, indicating *router drift*. (d) The left y-axis shows the normalized entropy, defined as the entropy divided by the maximum possible entropy over $n$ experts. Even after re-training the router on Task 1 while freezing experts from each stage, the recovered Task 1 accuracy drops across tasks and the routing entropy decreases, revealing *expert drift* beyond misrouting.

## 1. Introduction

Multimodal Large Language Models (MLLMs) (Bai et al., 2023; Liu et al., 2023; Zhu et al., 2023; Wan et al., 2025; Peng et al., 2026) have demonstrated impressive generalization capabilities through multimodal instruction tuning (Zhang et al., 2026; Tong et al., 2025) on large-scale datasets, enabling a model to perform a wide range of vision-language tasks (Radford et al., 2021; Dai et al., 2023; Guo et al., 2025b). However, in realistic scenarios, multimodal tasks (Hu & Singh, 2021; Yang et al., 2025b) are encountered sequentially, requiring MLLMs to expand their capabilities (Zhou et al., 2024b;a). In this multimodal continual instruction tuning (MCIT) (Chen et al., 2024) setting, MLLMs are required to continually master new task capabilities while preserving previously learned knowledge, which remains challenging due to catastrophic forgetting (Liu et al., 2025b; Wen et al., 2025; Xie et al., 2026).

To resist forgetting, recent works (Guo et al., 2025a; Huai

---
[*]Equal contribution [1]School of Artificial Intelligence, Nanjing University, China [2]State Key Laboratory of Novel Software Technology, Nanjing University, China. Correspondence to: Da-Wei Zhou <zhoudw@lamda.nju.edu.cn>.

*Proceedings of the 43rd International Conference on Machine Learning*, Seoul, South Korea. PMLR 306, 2026. Copyright 2026 by the author(s).

et al., 2025; Yang et al., 2025c; Yu et al., 2025) have increasingly explored Mixture-of-Experts (MoE) architectures (Jacobs et al., 1991) with LoRA (Hu et al., 2022) for MCIT, leveraging sparse expert routing and conditional computation to promote specialization across tasks (Qiao et al., 2024; Wang et al., 2025). Despite their intuitive appeal, these methods still exhibit substantial performance degradation on earlier tasks as training progresses (Li et al., 2025a).

To probe distinct sources of forgetting in MoE-based MCIT, we design diagnostic experiments in Fig. 1 by inserting MoE modules into the FFN layers of the MLLM. Consider an eight-task MCIT task (Chen et al., 2024), we save snapshots of the router and experts after each task, and reuse the test set of Task 1 to track routing behavior. Specifically, we feed Task 1 test samples into the router after training each subsequent task and compare the expert-activation distributions to the distribution just after learning Task 1. In Fig. 1a∼1c, we observe a progressively larger distribution shift: the activation pattern on Task 1 drifts away from its original snapshot, suggesting that previously seen inputs are increasingly reassigned to different experts. This instability is a direct symptom of *router drift*, which erodes the model's ability to reliably leverage prior task experts.

To further examine whether expert drift exists beyond router drift, we freeze the corresponding experts and re-train only the router on Task 1's training set after learning later tasks. We then evaluate on the Task 1 test set. As shown in Fig. 1d, despite using a router re-matched to Task 1, the accuracy of later expert snapshots still cannot recover to the Task 1 baseline and degrades as training proceeds, with particularly severe drops after Task 2 and Task 5. Meanwhile, the entropy of the re-trained router's outputs decreases over time, indicating a more peaked but increasingly constrained routing decision. These findings show that forgetting persists even under favorable routing, providing evidence of *expert drift*, *i.e.*, the experts themselves lose Task 1 functionality due to continual updates, rather than misrouting alone.

The above analyses imply that forgetting in MoE-based MCIT can arise from two coupled sources: (i) router drift, where inputs from old tasks are mismatched to wrong experts, resulting in inconsistent access to correct experts; (ii) expert drift, where experts themselves continue to change as they are reused across tasks, resulting in functional degradation on previously learned tasks.

In this paper, we propose StAbilized Mixture-of-Experts (SAME) for MCIT. To address router drift, SAME stabilizes expert selection via spectral-aware routing that decomposes update dynamics into task-relevant subspaces. To mitigate expert drift, we apply curvature-aware Riemannian scaling to regulate expert updates using historical input covariance in a rehearsal-free manner. Moreover, SAME introduces adaptive expert activation to freeze selected experts during

task training, reducing redundant computation and cross-task interference. Extensive experiments demonstrate that SAME consistently outperforms existing SOTA methods.

## 2. Related Work

**Multimodal Large Language Models.** The emergence of multimodal large language models (MLLMs) (Zhang et al., 2026; Touvron et al., 2023; Yang et al., 2025a; Du et al., 2026; Gong et al., 2026; Zhang et al., 2025a) has revolutionized vision-language understanding and generation. These models typically integrate a frozen vision encoder with a large language model (LLM) via cross-modal alignment mechanisms (Radford et al., 2021; Wen et al., 2025; Xie et al., 2026). Recent advances have significantly enhanced their capabilities in visual reasoning (Johnson et al., 2017; Zerroug et al., 2022), instruction following (Zhou et al., 2023), and generation (Feng et al., 2025). However, most existing MLLMs are trained in a static multi-task setting, ignoring the real-world requirement of continually arriving data stream (Shi et al., 2021; Bai et al., 2023).

**Continual Instruction Tuning for MLLMs.** As MLLMs are increasingly deployed in open-world settings, continual instruction tuning (Liu et al., 2023; Longpre et al., 2023; Liu et al., 2025b; Ning et al., 2025; Zhang et al., 2025b) without forgetting becomes essential. Existing methods mainly follow three complementary directions: replay-based strategies (Li et al., 2025b; Lee et al., 2025) that retain or synthesize prior image-text data to preserve past knowledge at the cost of storage or computation, cross-modal regularization-based methods (Zeng et al., 2025; Liu et al., 2025b) that constrain representation drift via alignment or parameter regularization under task shifts, and parameter-efficient adaptation-based approaches (Wang et al., 2025; Liu et al., 2025a) that update only a small set of lightweight task-specific modules while keeping the backbone frozen.

## 3. Preliminaries

**Multimodal continual instruction tuning (MCIT).** We consider an MLLM (Liu et al., 2023) consisting of a vision encoder, a multimodal projector, and a large language model. Let $\{D_1, D_2, \cdots, D_T\}$ denote the task sequence, where each task $D_t = \{(\mathbf{v}_i, \mathbf{q}_i, \mathbf{y}_i)\}_{i=1}^{n_t}$. $\mathbf{v}_i$ is an image, $\mathbf{q}$ is an instruction, and $\mathbf{y}$ is the target answer. We write $\tilde{\mathbf{v}}_i = \phi(\mathbf{v}_i) \in \mathbb{R}^{m \times d_v}$ for visual features extracted by a frozen vision encoder $\phi(\cdot)$ and $\mathbf{u}_i = \psi(\mathbf{q}_i) \in \mathbb{R}^{s \times d_u}$ for instruction token embeddings produced by the tokenizer and embedding layer $\psi(\cdot)$. A frozen projector $\pi(\cdot)$ maps visual features into the language embedding space, yielding $\mathbf{w}_i = \pi(\tilde{\mathbf{v}}_i) \in \mathbb{R}^{m \times d}$. The multimodal input sequence can be denoted by the concatenation $\mathbf{z}_i = [\mathbf{w}_i; \mathbf{u}_i] \in \mathbb{R}^{(m+s) \times d}$. Given a target response token sequence $\mathbf{y} = (y_1, \ldots, y_L)$,

the MLLM models the conditional distribution:

$$p_\theta(\mathbf{y}|\mathbf{z}) = \prod_{j=1}^{L} p_\theta(y_j|\mathbf{z}, \mathbf{y}_{<j}), \qquad (1)$$

where $\theta$ denotes trainable parameters. The optimization objective is to build a unified model that performs well on all tasks observed so far:

$$\theta_t^* = \arg\min_\theta \mathbb{E}_{(\mathbf{v},\mathbf{q},\mathbf{y})\sim\mathcal{D}_{\leq t}} \left[ -\sum_{j=1}^{L} \log p_\theta(\mathbf{y}_j|\mathbf{z}, \mathbf{y}_{<j}) \right],$$

where $\mathcal{D}_{\leq t}$ denotes the data distribution of all seen tasks.

**MoE with LoRA Experts.** Recent works often combine MoE with LoRA experts to enable parameter-efficient adaptation. These modules are added to a frozen backbone for conditional computation. In this paper, we focus on adding trainable parameters only to the FFN layers of the LLM (Wang et al., 2025; Zhu et al., 2025). For example, given an input $\mathbf{x} \in \mathbb{R}^d$ at layer $\ell$, we apply a gated mixture of LoRA updates to the frozen weights $\mathbf{W}_0$ with low-rank matrices $\mathbf{A}_i \in \mathbb{R}^{in \times r}$ and $\mathbf{B}_i \in \mathbb{R}^{r \times out}$:

$$\mathbf{h} = \mathbf{W}_0\mathbf{x} + \sum_{i=1}^{n} \omega_i \mathbf{W}_i \mathbf{x} = \mathbf{W}_0\mathbf{x} + \sum_{i=1}^{n} \omega_i \mathbf{B}_i \mathbf{A}_i \mathbf{x}, \quad (2)$$

where $\omega_i = \text{Softmax}(\mathbf{W}_G\mathbf{x})_i$ is the weight for $i$-th expert.

**Discussions.** While MoE with LoRA presents an avenue for MCIT, the paradigm in Eq. (2) can cause catastrophic forgetting from two sources: (i) as new tasks come, the router weight $\mathbf{W}_G$ can experience router drift, where the routing decisions $\omega_i$ become inconsistent over time, causing previously seen inputs to be assigned to different experts; (ii) the expert weight $\mathbf{W}_i$ can undergo expert drift, where repeated updates gradually degrade the expert's functionality on earlier tasks, leading to a loss of previously acquired knowledge. These issues are exacerbated as new tasks arrive, with the increasing diversity of inputs and the complexity of vision-language representations creating a high-dimensional routing space. In this space, even slight shifts in task distribution can lead to significant expert reassignment and destabilize the functionality of experts. Therefore, a routing mechanism capable of addressing both drifts is desired.

## 4. Method

To address the observed challenges, we introduce StAbilized Mixture-of-Experts (SAME) for scalable continual instruction tuning. SAME mitigates router drift via spectral-aware routing that updates routing weights in task-relevant subspaces. To control expert drift, we apply curvature-aware Riemannian scaling to preserve prior expert behaviors. Finally, SAME adopts adaptive expert activation to freeze selected experts at the task level, reducing redundant computation and cross-task interference.

### 4.1. Spectral-aware Routing

As discussed above, router drift arises when the router must extrapolate beyond its training distribution as new tasks arrive, causing previously seen inputs to be mapped to different experts over time. To address this, we propose to stabilize routing using spectral-aware consolidation. Specifically, we decompose the routing dynamics into orthogonal subspaces, updating only the directions vital for the current task while preserving those critical for previous tasks.

Let $\mathbf{W}_G^t$ denote the router weight of a layer for task $t$. During task $t$, we maintain an uncentered covariance $\mathbf{C}^t \approx \mathbb{E}_{\mathbf{x}\sim\mathcal{D}_{\leq t}} \mathbf{x}\mathbf{x}^\top$ of the hidden input distribution for the router along with updates:

$$\mathbf{C}^t = \frac{\alpha_{t-1}\mathbf{C}^{t-1} + n_t \hat{\mathbf{C}}^t}{\alpha_t}, \quad \alpha_t = \alpha_{t-1} + n_t, \quad (3)$$

where $n_t$ is the number of samples in task $t$ and $\hat{\mathbf{C}}^t$ is the sample covariance of the current task $t$, with initial values set as $\mathbf{C}^0 = \mathbf{0}$ and $\alpha_0 = 0$. However, storing the full matrix $\mathbf{C}^t \in \mathbb{R}^{d\times d}$ is prohibitively expensive in terms of memory. To address this, we simplify the storage by retaining only the first $k$ principal components, where $k$ is the smallest index such that the cumulative energy $\sum_{i=1}^{k} \sigma_i^2 / \sum_{i=1}^{d} \sigma_i^2 \geq \delta$ exceeds a preset threshold $\delta$. This ensures that we capture the most significant directions for gradient scaling while reducing memory usage. We then perform decomposition on $\mathbf{C}^t$ to identify high-energy and low-energy subspaces:

$$\mathbf{C}^t = \mathbf{U}\mathbf{\Sigma}\mathbf{V}^\top, \quad \mathbf{\Sigma} = \text{diag}(\sigma_1 \geq \cdots \geq \sigma_d). \quad (4)$$

It can further be separated into two orthogonal subspaces based on their importance for all seen tasks:

$$\mathbf{V}_{\|} = \mathbf{V}[:,:k], \quad \mathbf{V}_{\perp} = \mathbf{V}[:,k:d], \quad (5)$$

where $\mathbf{V}_{\|}$ represents directions important for the new task, while $\mathbf{V}_{\perp}$ captures directions primarily associated with old tasks, with minimal variance in the updated task distribution. We project the raw gradient $\Delta\mathbf{W}_G^t$ onto the important directions for the new task captured by $\mathbf{V}_{\|}$, which emphasizes updates along the directions essential for the current task, while retaining the critical components for previous tasks:

$$\Delta\mathbf{W}_{\|}^t = \Delta\mathbf{W}_G^t \mathbf{V}_{\|}\mathbf{V}_{\|}^\top. \quad (6)$$

While this projection focuses updates on the task-relevant subspace, it treats all directions within $V_{\|}$ equally. In practice, these directions can differ substantially in their relative importance for learning the new task. To address this, we propose to rescale the singular values. Specifically, for each singular value $\sigma_i$ of $\mathbf{V}_{\|}$, we compute a sliding window average $\hat{\sigma}_i$ as $\hat{\sigma}_i = \frac{1}{k}\sum_{j=i-k+1}^{i} \sigma_j$, which provides a smoothed estimate of the local context of each singular value [1]. We

---

[1] For indices near the boundary ($i < k$), we truncate the window to the available range, i.e., $\hat{\sigma}_i = \frac{1}{i}\sum_{j=1}^{i} \sigma_j$.

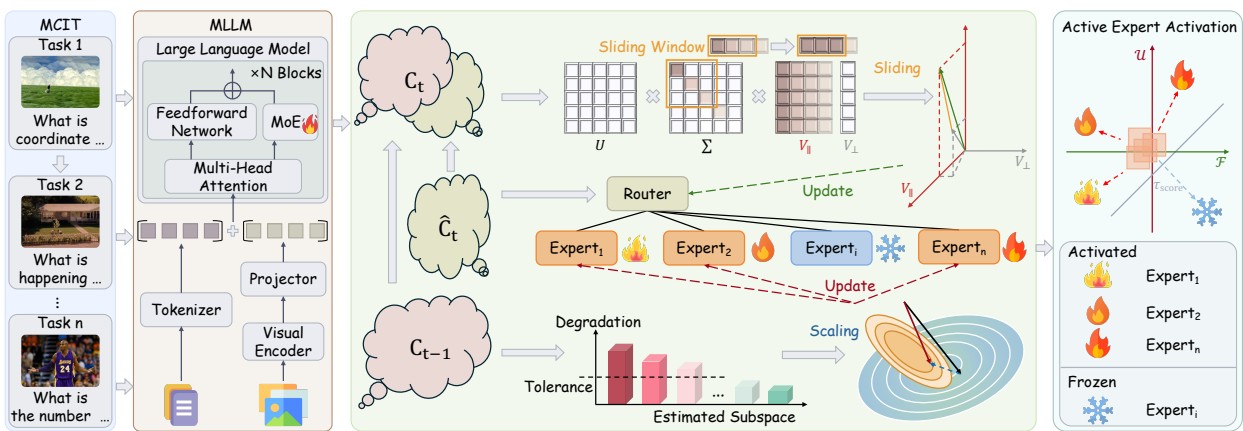

*Figure 2.* Overview of SAME. SAME stabilizes MoE adaptation by (i) tracking the router-input covariance and performing spectral-aware routing updates in task-relevant subspaces, (ii) applying curvature-aware scaling to bound expert degradation under historical input geometry, and (iii) using adaptive expert activation to freeze selected experts during each task.

then take a scaling function $g(\Sigma)$ to modulate updates to the router weights based on these smoothed values:

$$g(\Sigma) = \mathrm{diag}(\alpha_1\sigma_1, \alpha_2\sigma_2, \ldots, \alpha_r\sigma_r), \quad (7)$$

where $\alpha_i = 1/\hat{\sigma}_i$ adjusts the update based on the relative importance of each direction. This ensures that directions with smaller relative singular values, which are critical for maintaining old-task functionality, are updated less aggressively. Therefore, the update in Eq. (6) can be revised to:

$$\Delta\mathbf{W}_{\|}^t = \Delta\mathbf{W}_G^t\mathbf{V}_{\|}g(\Sigma)\mathbf{V}_{\|}^{\top}. \quad (8)$$

For the remaining directions associated with old-task knowledge, we consider projecting the raw gradient $\Delta\mathbf{W}_G^t$ onto the approximate null space using $\mathbf{V}_{\perp}$:

$$\Delta\mathbf{W}_{\perp}^t = \Delta\mathbf{W}_G^t\mathbf{V}_{\perp}\mathbf{V}_{\perp}^{\top}. \quad (9)$$

Since $\mathbf{C}^t \propto \mathbf{X}\mathbf{X}^{\top}$ for past router input distribution $\mathbf{X}$, the columns of $\mathbf{V}_{\perp}$ span directions with near-zero variance. As a result, $\mathbf{V}_{\perp}^{\top}\mathbf{X}^{\mathrm{old}} \approx \mathbf{0}$ for old–task features $\mathbf{X}^{\mathrm{old}}$:

$$\Delta\mathbf{W}_{\perp}^t\mathbf{X}^{\mathrm{old}} = \Delta\mathbf{W}_G^t\mathbf{V}_{\perp}\mathbf{V}_{\perp}^{\top}\mathbf{X}^{\mathrm{old}} \approx \mathbf{0}, \quad (10)$$

which ensures that updates to router weights preserve old-task predictions. More details are deferred to Appendix B.

To combine the updates to the router weights along the important directions for new tasks ($\Delta\mathbf{W}_{\|}^t$) and those associated with old-task knowledge ($\Delta\mathbf{W}_{\perp}^t$), we compute the final update as a weighted sum of the two components:

$$\Delta\mathbf{W}_G^t = \Delta\mathbf{W}_{\|}^t + \Delta\mathbf{W}_{\perp}^t. \quad (11)$$

This combined update ensures that the router's weight updates are focused on directions important for new tasks, while preserving the stability of old-task knowledge, thus effectively mitigating router drift.

**Discussions.** By decomposing routing updates into task-relevant and history-preserving subspaces, our spectral-aware routing stabilizes expert assignments across tasks and reduces router drift. This prevents unnecessary re-routing for previously seen input distribution while maintaining efficient adaptation to new tasks.

### 4.2. Curvature-aware Scaling

While spectral-aware routing stabilizes expert selection, continual instruction tuning can still cause the experts themselves to drift. This expert drift occurs when updates driven by new tasks overwrite expert functionalities that were critical to previous tasks, leading to irreversible degradation. To prevent such destructive interference, we regulate expert updates with a curvature-aware scaling rule that explicitly favors function preservation under historical inputs.

In a rehearsal-free setting, we cannot revisit past data to assess how much an expert has changed. Instead, we approximate the historical input geometry using the covariance $\mathbf{C}^{t-1}$ accumulated up to task $t-1$ via Eq. (3). For a LoRA expert $i$ with output contribution $\mathbf{h} = \mathbf{W}_i\mathbf{x}$, an update $\Delta\mathbf{W}_i$ induces a functional change $\Delta\mathbf{h}_i = \Delta\mathbf{W}_i\mathbf{x}$. We quantify degradation as the expected squared functional change on the historical input distribution:

$$\begin{aligned}\Delta_{\mathrm{degrad}} &\triangleq \mathbb{E}_{\mathbf{x}\sim\mathcal{D}_{<t}}\left[\|\Delta f_i(\mathbf{x})\|^2\right] \\ &= \mathbb{E}_{\mathbf{x}\sim\mathcal{D}_{<t}}\left[\|\Delta\mathbf{W}_i\mathbf{x}\|^2\right] = \mathrm{tr}\left(\Delta\mathbf{W}_i\mathbf{C}^{t-1}\Delta\mathbf{W}_i^{\top}\right),\end{aligned} \quad (12)$$

where $\mathrm{tr}(\cdot)$ denotes the trace of a matrix. This quantity penalizes updates that induce large output deviations along directions frequently observed in past tasks. Directly minimizing $\Delta_{\mathrm{degrad}}$ would overly constrain learning on the new task. Instead, we optimize current-task performance while explicitly bounding the permissible functional drift:

$$\min_{\Delta\mathbf{W}_i} \mathcal{L}(\Delta\mathbf{W}_i) + \lambda\max\left(0, \Delta_{\mathrm{degrad}} - \epsilon\right), \quad (13)$$

where $\epsilon$ defines the tolerance of functional deviation and $\lambda$ controls the strength of drift regularization. This formulation enforces stability only when the induced degradation exceeds the allowed budget, preserving plasticity for learning new tasks. This objective naturally leads to a Riemannian-scaled update under the metric induced by $\mathbf{C}^{t-1}$. Specifically, instead of using the Euclidean gradient, we precondition the update along the input geometry:

$$\Delta\mathbf{W}_i = -\eta\nabla_{\mathbf{W}_i}\mathcal{L}(\mathbf{C}^{t-1})^{-1}, \ \nabla_{\mathcal{M}}\mathcal{L} = \nabla_{\mathbf{W}_i}\mathcal{L}(\mathbf{C}^{t-1})^{-1}, \quad (14)$$

where $\nabla_{\mathcal{M}}\mathcal{L}$ denotes the Riemannian gradient of the loss on the manifold $\mathcal{M}$ equipped with the metric tensor $\mathbf{C}^{t-1}$ and $\eta$ is the learning rate. As shown in Eq. (14), we obtain the Riemannian gradient by scaling the Euclidean gradient $\nabla_{\mathbf{W}_i}\mathcal{L}$ with $(\mathbf{C}^{t-1})^{-1}$. Intuitively, this update downweights directions that correspond to high-variance historical features, preventing the expert from being significantly altered along dimensions that were heavily relied upon by previous tasks. However, directly inverting $\mathbf{C}^{t-1}$ is infeasible for large models. We therefore reuse the low-rank factors $(\mathbf{V}_k, \boldsymbol{\Sigma}_k)$ of $\mathbf{C}^{t-1}$ obtained in Eq. (4), and compute a numerically stable inverse using a damped pseudo-inverse:

$$(\mathbf{C}^{t-1})^{-1} \approx \mathbf{V}_k (\boldsymbol{\Sigma}_k + \mu\mathbf{I})^{-1} \mathbf{V}_k^\top + \frac{1}{\mu}\left(\mathbf{I} - \mathbf{V}_k\mathbf{V}_k^\top\right), \quad (15)$$

where $\mu > 0$ is a damping constant and $\mathbf{I}$ denotes the identity matrix. The first term performs a regularized inversion within the retained principal subspace, while the second term provides a well-conditioned default scaling in the orthogonal complement. This design avoids numerical instability caused by near-singular directions and enables drift-aware preconditioning with negligible memory overhead. More details are deferred to Appendix C.

**Discussions.** By measuring drift as functional deviation under historical input geometry and preconditioning expert updates accordingly, our method preserves previously acquired expert behaviors while maintaining sufficient capacity for new-task adaptation. This yields a scalable mechanism to mitigate expert drift throughout continual instruction tuning.

### 4.3. Adaptive Expert Activation

Conventional top-$k$ routing reduces inference cost by activating only a small subset of experts per token. However, in continual instruction tuning, such sample-level routing tends to scatter updates from a single task across many experts. This weakens knowledge compartmentalization and causes widespread parameter interference, where experts are repeatedly perturbed by new tasks and gradually lose functionalities acquired from earlier tasks. To mitigate this expert drift while improving training efficiency, we propose adaptive expert activation, a task-level mechanism that selectively freezes a subset of experts during training. Crucially,

freezing is only applied during the training of the current task: selected experts are temporarily frozen to stop their forward and backward propagation, and will be reactivated in subsequent tasks and at inference time.

Our goal is to freeze experts that (i) contribute little to the current task, yet (ii) encode valuable functionalities for historical tasks. We therefore rank experts using two complementary signals: *utilization* on the current task and *historical importance* accumulated from previous tasks.

If an expert is rarely activated while training task $t$, updating it provides little benefit for learning the current knowledge while incurring redundant computation overhead. We measure this using the running average routing weight. For each expert $i$, we maintain its *utilization* on the current task as

$$\mathcal{U}(i) \leftarrow \frac{n\mathcal{U}(i) + \sum_{\mathbf{x}\in\mathcal{B}}\omega_i(\mathbf{x})}{n + |\mathcal{B}|}, \quad n \leftarrow n + |\mathcal{B}|, \quad (16)$$

where $|\mathcal{B}|$ is the current batch size, and $\omega_i(\mathbf{x})$ denotes the routing weight assigned to expert $i$ for input $\mathbf{x}$.

However, *utilization* alone is insufficient: an expert may be active on the current task but still carry critical behaviors from previous tasks, and unconstrained updates may overwrite such functionalities. To estimate expert-level *historical importance* in a rehearsal-free manner, we adopt the trace of the Average Gradient Outer Product (AGOP) (Radhakrishnan et al., 2024) as a curvature-based sensitivity indicator. Directly computing AGOP is expensive, as it requires second-order curvature estimation through per-sample gradient outer products. Therefore, we use a lightweight proxy: for linear experts, the AGOP trace can be approximated by the routing-weighted input energy. We therefore maintain the following running estimate during task $t$:

$$\mathcal{F}^{\text{cur}}(i) \leftarrow \frac{n\mathcal{F}^{\text{cur}}(i) + \sum_{\mathbf{x}\in\mathcal{B}}\omega_i(\mathbf{x})\|\mathbf{x}\|^2}{n + |\mathcal{B}|}, \ n \leftarrow n + |\mathcal{B}|. \quad (17)$$

At the beginning of each task, we initialize $\mathcal{F}^{\text{cur}}(i) \leftarrow \mathcal{F}^{\text{pre}}(i)$ to carry forward *historical importance*, and after finishing task $t$, we update $\mathcal{F}^{\text{pre}}(i) \leftarrow \mathcal{F}^{\text{cur}}(i)$. Additional derivations are deferred to Appendix D.

Taken together, $\mathcal{U}(i)$ captures how much expert $i$ is needed for learning the current task, while $\mathcal{F}^{\text{pre}}(i)$ reflects its *historical importance* that should be preserved. To make these two signals comparable and derive a unified freezing criterion, we apply min-max normalization within each MoE layer, yielding $\tilde{\mathcal{U}}(i)$ and $\tilde{\mathcal{F}}^{\text{pre}}(i)$ and define an activation score:

$$\text{Score}(i) = \tilde{\mathcal{U}}(i) - \tilde{\mathcal{F}}^{\text{pre}}(i). \quad (18)$$

We temporarily freeze experts with $\text{Score}(i) < \tau_{\text{score}}$ during training of the current task, where $\tau_{\text{score}}$ controls the aggressiveness of freezing. This rule prioritizes freezing experts

that are *redundant* for learning task $t$ ($\tilde{\mathcal{U}} \downarrow$) yet are *valuable to preserve* for earlier tasks ($\tilde{\mathcal{F}}^{\mathrm{pre}} \uparrow$), thereby reducing unnecessary updates and protecting historical behaviors.

**Discussions.** By freezing experts that are unhelpful for the current task yet important to preserve, adaptive expert activation reduces redundant training computation and prevents unnecessary parameter drift. This encourages task-specific updates to concentrate on a smaller subset of experts, strengthening knowledge compartmentalization and mitigating interference across tasks. Together with the routing stabilization in Sec. 4.1, this yields more stable expert specialization throughout continual instruction tuning.

### 4.4. Summary of SAME

SAME addresses two key challenges in MCIT: router drift and expert drift. We stabilize routing through spectral-aware updates to maintain consistent expert selection across tasks, and regulate expert adaptation via curvature-aware Riemannian scaling to preserve previously learned behaviors. In addition, SAME employs adaptive expert activation to freeze selected experts during task training, reducing redundant computation and cross-task interference. The complete training procedure is summarized in Appendix A.

## 5. Experiments

### 5.1. Implementation Details

**Datasets.** To evaluate our method, we conduct experiments across three benchmarks: CoIN (Chen et al., 2024), UCIT (Guo et al., 2025a), and our newly proposed TriGap. CoIN comprises eight sequential tasks: ScienceQA (Lu et al., 2022), TextVQA (Singh et al., 2019), ImageNet (Deng et al., 2009), GQA (Hudson & Manning, 2019), VizWiz (Gurari et al., 2018), REC (Kazemzadeh et al., 2014), VQAv2 (Goyal et al., 2017), and OCR-VQA (Mishra et al., 2019). UCIT includes six tasks: ArxivQA (Li et al., 2024), CLEVR-Math (Lindström & Abraham, 2022), IconQA (Lu et al., 2021), ImageNet-R (Hendrycks et al., 2021), VizWiz-Caption (Gurari et al., 2018), and Flickr30k (Plummer et al., 2015).

While CoIN and UCIT cover standard MCIT settings, they typically feature short task sequences, limited domain variation, and balanced data scales. To stress-test model stability and adaptability under more realistic conditions, we introduce **TriGap** as a newly constructed benchmark. TriGap is designed around three practical dimensions that distinguish it from existing benchmarks: (1) *Extended Sequence & Instruction Gap*: We extend the horizon to 10 sequential tasks and employ diverse instruction formats to evaluate adaptive prompt understanding. (2) *Domain Span Gap*: Unlike prior benchmarks that focus primarily on general visual recognition and QA, TriGap incorporates specialized

domains such as scientific reasoning, remote sensing, and complex document/chart understanding, testing the model's ability to acquire heterogeneous capabilities across distinct fields. (3) *Dataset Scale Gap*: Per-task data sizes vary substantially (10k–40k samples), simulating realistic data imbalance to evaluate robustness against catastrophic forgetting under non-uniform streams. With over 250k training instances in total, TriGap provides a structured evaluation suite for studying multimodal continual instruction tuning under realistic sequence, domain, and scale shifts. In addition, TriGap is constructed from datasets absent from LLaVA's training mixture (Liu et al., 2023), eliminating potential information leakage and enabling cleaner MCIT evaluation. TriGap integrates ten diverse tasks, including PMCVQA (Zhang et al., 2023), DocVQA (Mathew et al., 2021), ChartQA (Masry et al., 2022), IconQA (Lu et al., 2021), InfographicVQA (Mathew et al., 2022), ArxivQA (Li et al., 2024), Roadside (Guan et al., 2026), ChemVQA (Sabando et al., 2020), FloodNetVQA (Rahnemoonfar et al., 2021), and CLEVR (Johnson et al., 2017).

**Compared Methods.** We compare our method with state-of-the-art methods, including MoELoRA (Chen et al., 2024), Continual LLaVA (Cao et al., 2024), ModalPrompt (Zeng et al., 2025), SEFE (Chen et al., 2025), ProgLoRA (Yu et al., 2025), LLaVA-CMoE (Zhao et al., 2025), CL-MoE (Huai et al., 2025) and HiDe-LLaVA (Guo et al., 2025a).

**Training Details.** All experiments are conducted on 8 NVIDIA RTX 5090 GPUs. We follow Prism (Tang et al., 2026) to conduct all experiments. Following Chen et al. (2024), we use LLaVA-v1.5-7B (Liu et al., 2023) as the backbone MLLM and CLIP-L/14-336 (Radford et al., 2021) to extract visual and textual features. Following Zhu et al. (2025), we only insert LoRA modules into the FFN layers of the language model, set the LoRA rank to 8. We train each task for 1 epoch with a warm-up ratio of 0.03. Only LoRA modules are trainable, with a learning rate of $2e^{-4}$ using a cosine decay schedule. We use a batch size of 6 for all methods. We provide the detailed hyperparameter configurations and sensitivity analyses in Appendix F.

**Evaluation Metrics.** Following Zhou et al. (2024b); Chen et al. (2024), we denote by $\mathcal{A}_{s,t}$ the performance on task $s$ evaluated after training up to task $t$, with $T$ total tasks. We summarize the average final performance by $\bar{\mathcal{A}} = \frac{1}{T} \sum_{s=1}^{T} \mathcal{A}_{s,T}$.

### 5.2. Benchmark Comparison and Ablation

**Benchmark Comparison.** We evaluate SAME against baselines on TriGap, CoIN, and UCIT (Tab. 1, 2, 3). On the more challenging TriGap benchmark, SAME reaches an average accuracy of 46.53%, outperforming MoELoRA by +2.08%. The gain is particularly clear on tasks with distinct visual and reasoning demands, such as DocVQA (43.87%) and

*Table 1.* Performance on the TriGap benchmark. Results for all baseline methods are reproduced under the same experimental setup and backbone. The best and second-best results are highlighted in **bold** and underline, respectively.

| Methods | PMCVQA | DocVQA | ChartQA | IconQA | InfographicVQA | ArxivQA | Roadside | ChemVQA | FloodNetVQA | CLEVR | Average |
|---|---|---|---|---|---|---|---|---|---|---|---|
| Zero-shot | 35.40 | 12.68 | 9.36 | 19.27 | 5.06 | 53.77 | 7.40 | 5.30 | 47.41 | 20.37 | 21.60 |
| FT-LoRA | 34.20 | 23.32 | 9.84 | 37.07 | 23.53 | 83.83 | 7.00 | 12.70 | 80.31 | 60.27 | 37.21 |
| Replay-LoRA | 33.70 | 33.95 | 14.00 | 46.67 | 28.97 | 75.57 | 9.40 | 15.90 | 73.81 | 58.80 | 39.08 |
| MoE-LoRA (Chen et al., 2024) | 39.03 | 37.49 | 12.44 | 43.43 | 35.17 | 90.90 | 7.93 | 20.70 | 90.41 | 67.00 | 44.45 |
| HiDe-LLaVA (Guo et al., 2025a) | 37.00 | 33.20 | 10.52 | 41.97 | 24.09 | 79.20 | 7.73 | 11.17 | 57.39 | 23.00 | 32.53 |
| ModalPrompt (Zeng et al., 2025) | 38.23 | 38.23 | 11.92 | 44.73 | 37.37 | 84.47 | 10.13 | 12.43 | 71.52 | 52.50 | 40.15 |
| CL-MoE (Huai et al., 2025) | 40.53 | 36.79 | 13.72 | 52.70 | 32.27 | 93.00 | 7.77 | 18.33 | 80.09 | 65.90 | 44.11 |
| SAME (Ours) | 41.60 | 43.87 | 17.56 | 64.03 | 39.57 | 90.46 | 10.83 | 21.77 | 81.09 | 54.50 | 46.53 |

*Table 2.* Average performance on the CoIN benchmark. Results for all baseline methods are directly adopted from their original publications. The best and second-best results are highlighted in **bold** and underline, respectively.

| Methods | ScienceQA | TextVQA | ImageNet | GQA | VizWiz | REC | VQAv2 | OCR-VQA | Accuracy |
|---|---|---|---|---|---|---|---|---|---|
| MoELoRA (Chen et al., 2024) | 62.02 | 52.05 | 37.21 | 53.12 | 43.32 | 33.22 | 57.92 | 65.75 | 50.58 |
| Continual LLaVA (Cao et al., 2024) | 58.67 | 49.99 | 57.66 | 62.53 | 42.32 | 16.25 | 64.33 | 74.91 | 53.33 |
| ModalPrompt (Zeng et al., 2025) | 68.42 | 56.40 | 41.13 | 61.11 | 50.13 | 36.69 | 66.90 | 59.68 | 55.06 |
| SEFE (Chen et al., 2025) | 75.35 | 58.66 | 83.10 | 54.25 | 48.85 | 16.75 | 65.35 | 66.25 | 58.57 |
| ProgLoRA (Yu et al., 2025) | 74.84 | 51.83 | 83.90 | 49.93 | 53.87 | 31.19 | 62.71 | 64.44 | 59.09 |
| LLaVA-CMoE (Zhao et al., 2025) | 77.55 | 58.17 | 94.50 | 48.91 | 55.45 | 23.40 | 56.40 | 59.44 | 59.23 |
| HiDe-LLaVA (Guo et al., 2025a) | 73.20 | 56.92 | 69.28 | 61.33 | 50.76 | 59.18 | 67.12 | 64.76 | 63.95 |
| SAME (Ours) | 78.35 | 60.69 | 90.21 | 61.70 | 54.13 | 59.87 | 66.04 | 63.59 | 66.82 |

FloodNet (81.09%), suggesting that SAME is better suited to long task sequences with heterogeneous instruction formats and non-uniform data scales. On CoIN and UCIT, SAME also maintains strong performance, with average accuracies of 66.82% and 67.12%, respectively. These consistent gains reflect the complementary roles of our three designs: spectral-aware routing reduces assignment drift by keeping historical samples routed to compatible experts, curvature-aware scaling preserves expert functionality along history-sensitive directions, and adaptive expert activation avoids updating experts that are less relevant to the current task. Together, these mechanisms reduce destructive updates and cross-task interference, leading to more reliable retention on both early and long-unseen tasks across benchmarks.

**Ablation Study.** To disentangle the contribution of each component, we conduct an ablation study as summarized in Tab. 4. **Baseline** denotes MoELoRA, which updates routers and experts without explicit constraints and thus suffers from unstable routing and expert overwriting. Adding **w/ Router** introduces spectral-aware routing, which stabilizes expert assignment by constraining router updates in history-preserving subspaces. The improvement confirms that consistent routing is essential for preventing old samples from being reassigned to inappropriate experts. Building on this, **w/ Expert** incorporates curvature-aware scaling to regulate expert updates under historical input geometry, thereby reducing expert drift along directions important to previous tasks. This brings clear gains in knowledge-intensive and format-sensitive tasks such as ScienceQA, where overwritten behaviors easily cause forgetting. Finally, **w/ Activation** introduces adaptive expert activation to temporarily freeze

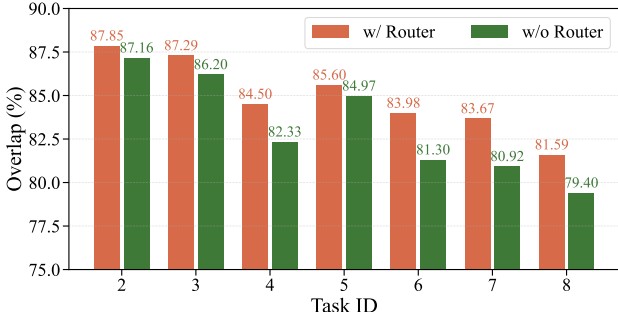

*Figure 3.* Impact of spectral-aware routing. Adding the spectral-aware routing strategy enables more consistent expert selection.

historically important but currently less useful experts, further reducing redundant updates and cross-task interference. Together, these results show that router stabilization, expert preservation, and adaptive activation provide complementary gains and jointly improve the robustness of SAME.

### 5.3. Further Analysis

**Impact of Spectral-aware Routing.** To assess whether spectral-aware routing mitigates router drift, we track how the router's output distribution on the Task 1 test set evolves over the course of continual tuning. Concretely, we record the routing distribution produced immediately after training Task1, and then re-evaluate the router on the same Task1 inputs after each subsequent task $t$. In Fig. 3, **w/o Router** denotes vanilla MoELoRA, which updates the router without constraints; its routing distribution drifts steadily as $t$ increases, implying that Task 1 samples are progressively reassigned to different experts. In contrast, **w/ Router** in-

*Table 3.* Performance on the UCIT benchmark. The best and second-best results are highlighted in **bold** and underline, respectively.

| Methods | ImageNet-R | ArxivQA | Vizcap | IconQA | CLEVR | Flickr30k | Average |
|---|---|---|---|---|---|---|---|
| Zero-shot | 18.88 | 52.62 | 38.75 | 21.25 | 21.12 | 41.44 | 32.34 |
| FT-LoRA | 29.33 | 55.30 | 45.51 | 26.13 | 13.07 | 58.07 | 37.90 |
| MoE-LoRA (Chen et al., 2024) | 49.87 | 77.63 | 43.65 | 46.40 | 36.47 | **58.34** | 52.06 |
| Replay-LoRA | 76.93 | 87.07 | **54.31** | 56.43 | 36.40 | 55.94 | 61.18 |
| CL-MoE (Huai et al., 2025) | 64.12 | 78.38 | 44.83 | 62.00 | 50.75 | 58.06 | 59.69 |
| HiDe-LLaVA (Guo et al., 2025a) | 80.50 | 89.83 | 48.78 | 62.90 | 47.97 | 55.15 | 64.19 |
| ModalPrompt (Zeng et al., 2025) | 81.50 | 90.62 | 50.17 | 62.34 | 51.41 | 57.09 | 65.52 |
| SAME (Ours) | **83.83** | **91.40** | 51.33 | **65.27** | **53.50** | 57.43 | **67.12** |

*Table 4.* Ablation studies of different components for SAME.

| Datasets | Baseline | w/ Router | w/ Expert | w/ Activation |
|---|---|---|---|---|
| ScienceQA | 62.02 | 71.44 | **80.29** | 78.35 |
| TextVQA | 52.05 | 59.64 | **66.85** | 60.69 |
| ImageNet | 37.21 | 70.54 | 84.49 | **90.21** |
| GQA | 53.12 | 58.99 | **62.20** | 61.70 |
| VizWiz | 43.32 | 48.06 | 52.73 | **54.53** |
| REC | 33.22 | 54.64 | 56.19 | **59.87** |
| VQAv2 | 57.92 | 64.68 | 61.21 | **66.04** |
| OCR-VQA | **65.75** | 62.64 | 63.19 | 63.59 |
| Accuracy ↑ | 50.58 | 61.32 | 65.89 | **66.82** |

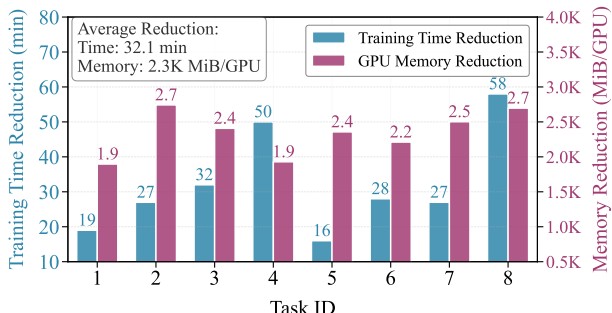

*Figure 5.* Impact of adaptive expert activation on training efficiency. By freezing low-utility yet historically important experts during each task, our method reduces per-task training time and GPU memory footprint across continual instruction tuning tasks.

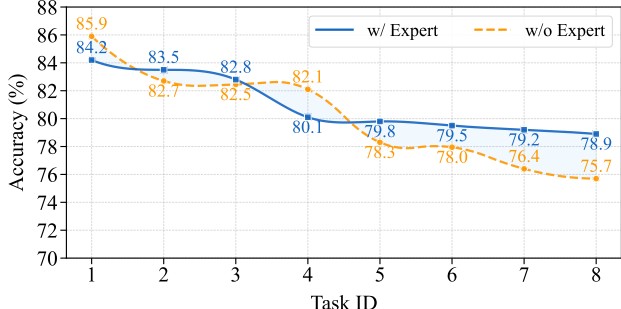

*Figure 4.* Impact of curvature-aware scaling. Adding curvature-aware scaling improves re-routing accuracy on Task 1, indicating stronger preservation of early-task expert functionality.

corporates our spectral-aware routing and exhibits markedly smaller distribution shift, indicating more consistent expert selection over time. This improved routing stability aligns with the stronger retention we observe on long-unseen tasks.

**Impact of Curvature-aware Scaling.** To further attribute forgetting to expert drift rather than router drift, we extend the diagnostic protocol in Fig. 1 and report the results in Fig. 4. We start from the same training setup with spectral-aware routing enabled, and only toggle curvature-aware scaling: **w/ Expert** applies the Riemannian preconditioning in Sec. 4.2, while **w/o Expert** performs standard expert updates. After completing each task $t$, we freeze the corresponding expert snapshot and then *re-train only the router* on Task 1 data before evaluating on the Task 1 test set. This re-routing protocol largely removes misrouting as a confounder, so the remaining accuracy reflects how much Task 1 functionality is still encoded in the experts.

Fig. 4 shows that enabling curvature-aware scaling consistently improves the recoverability of Task 1 performance

from later expert snapshots. While both variants gradually degrade as the training sequence grows, **w/ Expert** exhibits a markedly slower decline and maintains a larger margin in the later stages (Tasks 5–8), where cumulative interference is strongest. Qualitatively, this indicates that curvature-aware scaling suppresses updates along historically high-variance directions under $\mathbf{C}^{t-1}$, reducing destructive overwriting of features that were frequently used by earlier tasks. As a result, even after multiple rounds of continual instruction tuning, the experts retain more of the behaviors needed for Task 1, and the re-trained router can more effectively recover the original routing-to-function mapping.

**Impact of Adaptive Expert Activation.** To quantify the computational benefits of our adaptive expert activation in Sec. 4.3, we report the per-task reduction in training time and GPU memory footprint after enabling this module (on top of spectral-aware routing and curvature-aware scaling) in Fig. 5. By temporarily freezing a subset of experts during each task and thereby skipping their forward/backward computation, the training pipeline incurs substantially less backpropagation overhead and requires fewer activations to be stored. Under our surveillance, adaptive expert activation yields consistent savings throughout the sequence, reducing training time by 32.1 minutes per task on average and lowering GPU memory usage by 2.3 K MiB/GPU on average. The speedup is particularly pronounced on Task 4 and Task 8, where the time reduction reaches 50 and 58 minutes, respectively, indicating that task-level freezing becomes more beneficial as the model accumulates more

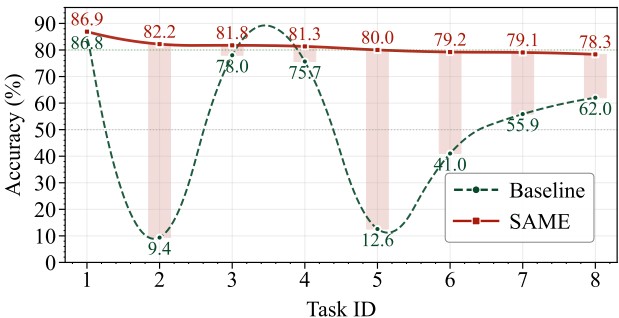

*Figure 6.* Mitigating formatting-induced forgetting with SAME. SAME avoids the recurring drop–rebound pattern on ScienceQA by preserving task-specific output formatting across tasks.

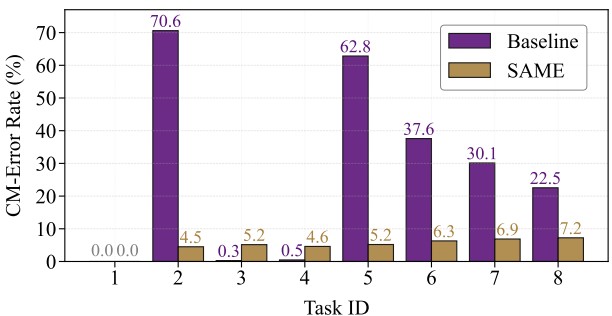

*Figure 7.* Case mismatch error rate on ScienceQA. After completing each task, we evaluate SAME and the baseline on the test set of ScienceQA and report the fraction of predictions that are semantically correct but incorrectly formatted in lowercase.

experts and routing becomes more selective. Meanwhile, the memory reduction remains stable across tasks (roughly 1.9–2.7 K MiB/GPU), confirming that freezing effectively alleviates activation storage pressure during training.

**Formatting-Induced Forgetting.** To further analyze performance degradation in continual instruction tuning, we take MoELoRA as our baseline and evaluate it on the ScienceQA test set after each task on CoIN. As shown in Fig. 6, we observe a striking and recurring non-monotonic forgetting pattern. ScienceQA accuracy drops sharply after Task 2 (TextVQA), rebounds unexpectedly after Task 3 (ImageNet), and then declines. Similar "drop-rebound" cycles recur around Task 5, indicating a systematic vulnerability rather than random fluctuation.

As shown in Fig. 7, error analysis reveals that the initial collapse is largely formatting-driven. After Task 2, 70.6% of predictions that are semantically correct are marked wrong solely due to letter casing: the model outputs lowercase (*e.g.*, "a") while ScienceQA requires uppercase (*e.g.*, "A"). This points to a distribution shift in answer formatting that TextVQA annotations are predominantly lowercase, and indicates that shared experts drift toward the new convention, overwriting the case-sensitive behavior acquired in Task 1.

The rebound after Task 3 further supports this explanation.

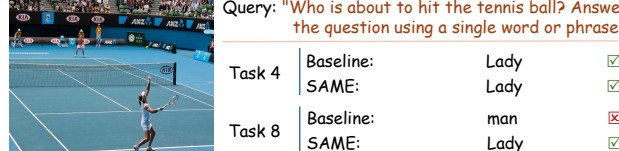

*Figure 8.* Qualitative comparison of prediction stability. SAME better preserves task-appropriate outputs than the Baseline.

ImageNet labels often follow a more capitalized style (*e.g.*, "Dog", "Golden retriever"), which nudges the model back toward an uppercase-compatible format, partially restoring ScienceQA scores without necessarily improving semantic competence. The same mechanism recurs after Task 5 (VizWiz), whose answers are again largely lowercase, triggering another sharp ScienceQA drop.

Overall, these results show that Baseline is highly susceptible to format drift: experts adapt to the current task's annotation style and inadvertently overwrite previously learned formatting conventions. In contrast, SAME remains stable across the sequence by curbing expert drift (via curvature-aware scaling) and reducing unnecessary expert updates (via adaptive expert activation), preserving both semantic competence and task-specific output format.

**Example Results.** We take MoELoRA as the Baseline and inspect predictions on earlier tasks at two checkpoints: right after a task is learned and after finishing the full training. Fig. 8 shows that SAME better preserves task-appropriate outputs under continual tuning. In the example, both methods predict "Lady" after Task 4 (GQA), but after training up to Task 8 the Baseline drifts to "man" while SAME remains consistent with "Lady", indicating stronger resistance to cross-task interference and better prediction stability.

## 6. Conclusion

In this paper, we study how to equip MLLMs with the ability to continually follow new user instructions under sequential training. We identify two key sources of forgetting in MoE-based continual instruction tuning: router drift and expert drift. To address these issues, we stabilize expert selection, limit destructive expert updates, and introduce adaptive expert activation that freezes selected experts during each task to reduce redundant computation and cross-task interference. Extensive experiments on benchmark datasets show that SAME consistently improves both retention and accuracy across diverse vision-language tasks while preserving training efficiency, making it an effective solution for MCIT.

**Limitations and Future Work.** While SAME is effective for rehearsal-free MCIT, further improving robustness remains important when task boundaries are ambiguous and input formats vary. Future work will explore tighter coupling between inference-time routing and drift control.

## Acknowledgments

This work was supported in part by the Basic Research Program of Jiangsu (BK20251251), NSFC (62506160, 62476123, 62522605, 62376118), JSTJ-2025-147, Fundamental and Interdisciplinary Disciplines Breakthrough Plan of the Ministry of Education of China (No. JYB2025XDXM118), the 111 Center (No. B26023), and the Collaborative Innovation Center of Novel Software Technology and Industrialization.

## Impact Statement

This paper presents work whose goal is to advance the field of Machine Learning. There are many potential societal consequences of our work, none of which we feel must be specifically highlighted here.

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

# A. Pseudocode of SAME

We summarize the overall procedure of SAME in Algorithm 1 and Algorithm 2. The training algorithm integrates spectral-aware routing, curvature-aware scaling, and adaptive expert activation, while inference uses the learned router and experts without freezing.

---

**Algorithm 1** Training of SAME for Continual Instruction Tuning

---

**Require:** Task stream $\{\mathcal{D}_t\}_{t=1}^T$, MoE model with router $\mathbf{W}_G$ and experts $\{\mathbf{W}_i\}$
**Require:** Hyperparameters: energy threshold $\delta$, damping $\mu$, drift control $(\epsilon, \lambda)$, freezing threshold $\tau_{\text{score}}$
**Ensure:** Updated router $\mathbf{W}_G$ and experts $\{\mathbf{W}_i\}$
 1: Initialize router covariance states: $\mathbf{C}^0 \leftarrow \mathbf{0}$, $\alpha_0 \leftarrow 0$
 2: Initialize expert importance buffer: $\mathcal{F}^{\text{pre}}(i) \leftarrow 0$ for all experts $i$
 3: **for** $t = 1$ to $T$ **do**
 4:     Reset per-task counters: $n \leftarrow 0$, $\mathcal{U}(i) \leftarrow 0$ for all experts $i$
 5:     Initialize $\mathcal{F}^{\text{cur}}(i) \leftarrow \mathcal{F}^{\text{pre}}(i)$ for all experts $i$
 6:     **for** each mini-batch $\mathcal{B} \subset \mathcal{D}_t$ **do**
 7:         **Forward routing:** compute routing weights $\{\omega_i(\mathbf{x})\}$ for $\mathbf{x} \in \mathcal{B}$
 8:         **Update covariance (router input):** update $\mathbf{C}^t$ using Eq. (3)
 9:         Retain top-$k$ principal components by the energy criterion $\delta$, and obtain $(\mathbf{V}_\parallel, \mathbf{V}_\perp)$ (Eq. (4)–Eq. (5))
10:         **Adaptive expert activation:**
11:         Update expert utilization $\mathcal{U}(i)$ on task $t$ using Eq. (16)
12:         Update expert historical-importance proxy $\mathcal{F}^{\text{cur}}(i)$ using Eq. (17)
13:         Compute activation score $\text{Score}(i)$ using Eq. (18)
14:         Freeze experts with $\text{Score}(i) < \tau_{\text{score}}$ for the current task
15:         **Spectral-aware routing update:**
16:         Compute router gradient $\Delta\mathbf{W}_G^t$ on $\mathcal{B}$
17:         Form direction-aware update in $\mathbf{V}_\parallel$ (Eq. (6), Eq. (8))
18:         Form history-preserving update in $\mathbf{V}_\perp$ (Eq. (9))
19:         Combine and apply router update (Eq. (11))
20:         **Curvature-aware scaling for experts:**
21:         Optimize the drift-aware objective in Eq. (13)
22:         Precondition expert gradients using $(\mathbf{C}^{t-1})^{-1}$ (Eq. (14))
23:         Approximate $(\mathbf{C}^{t-1})^{-1}$ via damped low-rank pseudo-inverse (Eq. (15))
24:         Update only *unfrozen* experts $\{\mathbf{W}_i\}$ with the scaled gradients
25:     **end for**
26:     Commit historical importance: $\mathcal{F}^{\text{pre}}(i) \leftarrow \mathcal{F}^{\text{cur}}(i)$ for all experts $i$
27: **end for**

---

**Algorithm 2** Inference of SAME (Routing + Prediction)

---

**Require:** Test input $\mathbf{x}$, trained MoE model (router $\mathbf{W}_G$, experts $\{\mathbf{W}_i\}$)
**Ensure:** Prediction $\hat{y}$
 1: **Routing:** compute router scores and routing weights $\{\omega_i(\mathbf{x})\}$ using $\mathbf{W}_G$
 2: Select top-$k$ experts according to routing weights
 3: **Expert aggregation:** compute expert outputs on $\mathbf{x}$ and aggregate them using $\{\omega_i(\mathbf{x})\}$
 4: Output final prediction $\hat{y}$ from the aggregated expert response

---

# B. Projection of Historical Inputs onto the Null Space

Let $\mathbf{C}^t = \mathbb{E}_{\mathbf{x} \sim \mathcal{D}_{\leq t}}[\mathbf{x}\mathbf{x}^\top] \in \mathbb{R}^{d \times d}$ denote the uncentered second-moment matrix of the hidden input distribution up to task $t$, where $\mathcal{D}_{\leq t}$ represents the union of data distributions from all observed tasks. Unlike conventional covariance matrices, $\mathbf{C}^t$ captures raw energy distribution without centering, making it particularly suitable for transformer representations that typically exhibit near-zero mean after normalization layers.

Performing singular value decomposition on $\mathbf{C}^t$ yields:

$$\mathbf{C}^t = \sum_{i=1}^d \sigma_i^2 \mathbf{v}_i \mathbf{v}_i^\top = \mathbf{V}\mathbf{\Sigma}\mathbf{V}^\top,$$

where $\sigma_1^2 \geq \sigma_2^2 \geq \cdots \geq \sigma_d^2 \geq 0$ are the eigenvalues (squared singular values) arranged in descending order, and $\mathbf{v}_i$ are the corresponding orthonormal eigenvectors forming the columns of $\mathbf{V} \in \mathbb{R}^{d \times d}$. The eigenvalue $\sigma_i^2$ quantifies the average energy of the input distribution along direction $\mathbf{v}_i$:

$$\sigma_i^2 = \mathbb{E}_{\mathbf{x} \sim \mathcal{D}_{\leq t}}[(\mathbf{v}_i^\top \mathbf{x})^2].$$

We partition the eigenvectors based on their spectral energy contribution. Let $r$ be the smallest index such that the cumulative energy ratio exceeds a threshold $\delta \in (0, 1)$:

$$\frac{\sum_{k=1}^r \sigma_k^2}{\sum_{k=1}^d \sigma_k^2} \geq \delta.$$

This defines two orthogonal subspaces:

(i) The *signal subspace* $\mathcal{S}_\| = \mathrm{span}(\mathbf{V}_\|)$, where $\mathbf{V}_\| = [\mathbf{v}_1, \ldots, \mathbf{v}_r] \in \mathbb{R}^{d \times r}$ contains eigenvectors with significant energy contributions

(ii) The *approximate null space* $\mathcal{S}_\perp = \mathrm{span}(\mathbf{V}_\perp)$, where $\mathbf{V}_\perp = [\mathbf{v}_{r+1}, \ldots, \mathbf{v}_d] \in \mathbb{R}^{d \times (d-r)}$ contains eigenvectors with negligible energy

For any historical input $\mathbf{x}^{\mathrm{old}} \sim \mathcal{D}_{<t}$ from previous tasks, its expected squared projection onto the null space is:

$$\mathbb{E}[\|\mathbf{V}_\perp^\top \mathbf{x}^{\mathrm{old}}\|^2] = \mathbb{E}[\mathrm{tr}(\mathbf{V}_\perp^\top \mathbf{x}^{\mathrm{old}} \mathbf{x}^{\mathrm{old}\top} \mathbf{V}_\perp)] = \mathrm{tr}(\mathbf{V}_\perp^\top \mathbb{E}[\mathbf{x}^{\mathrm{old}} \mathbf{x}^{\mathrm{old}\top}] \mathbf{V}_\perp).$$

Since historical inputs constitute part of the distribution used to construct $\mathbf{C}^t$, and given the energy threshold criterion, we have:

$$\mathbb{E}[\|\mathbf{V}_\perp^\top \mathbf{x}^{\mathrm{old}}\|^2] = \sum_{k=r+1}^d \sigma_k^2 \leq (1-\delta)\sum_{k=1}^d \sigma_k^2.$$

By construction of the threshold $\delta$, the right-hand side is bounded by a small constant $\epsilon > 0$, yielding:

$$\mathbb{E}[\|\mathbf{V}_\perp^\top \mathbf{x}^{\mathrm{old}}\|^2] \leq \epsilon.$$

For practical implementations where $\delta$ is selected sufficiently close to 1, this bound ensures that:

$$\mathbf{V}_\perp^\top \mathbf{x}^{\mathrm{old}} \approx \mathbf{0}.$$

This property is critical for router stability: when weight updates are confined to the null space $\mathcal{S}_\perp$, their effect on historical inputs vanishes asymptotically. Formally, for any update $\Delta\mathbf{W}_\perp = \Delta\mathbf{W}_G \mathbf{V}_\perp \mathbf{V}_\perp^\top$, the induced change in routing logits satisfies:

$$\|\Delta\mathbf{W}_\perp \mathbf{x}^{\mathrm{old}}\| = \|\Delta\mathbf{W}_G \mathbf{V}_\perp (\mathbf{V}_\perp^\top \mathbf{x}^{\mathrm{old}})\| \leq \|\Delta\mathbf{W}_G\| \cdot \|\mathbf{V}_\perp\| \cdot \|\mathbf{V}_\perp^\top \mathbf{x}^{\mathrm{old}}\| \approx 0,$$

thus preserving routing decisions for previously learned tasks while allowing adaptation in the signal subspace for new knowledge acquisition.

## C. Derivation of Curvature-aware Scaling via Riemannian Gradient Descent

We provide a rigorous derivation of the curvature-aware scaling rule from the constrained optimization objective with hinge-penalty regularization. The derivation proceeds through four stages: formalizing functional degradation as a quadratic constraint, establishing equivalence between the hinge-penalty formulation and its Lagrangian dual, deriving the Riemannian gradient update via first-order approximation, and justifying the dynamic coupling between regularization strength and learning rate scheduling.

### C.1. Problem Formulation

Consider a LoRA expert with trainable weight matrix $\mathbf{W}_i \in \mathbb{R}^{d_{\text{out}} \times d_{\text{in}}}$ at task $t$. Let $\Delta \mathbf{W}_i$ denote the parameter update during training on task $t$. The functional degradation induced on historical tasks $\mathcal{D}_{<t}$ is defined as the expected squared output deviation:

$$\Delta_{\text{degrad}} \triangleq \mathbb{E}_{\mathbf{x} \sim \mathcal{D}_{<t}} \big[ \|\Delta \mathbf{W}_i \mathbf{x}\|^2 \big]. \tag{19}$$

Using the uncentered second-moment matrix $\mathbf{C}^{t-1} = \mathbb{E}_{\mathbf{x} \sim \mathcal{D}_{<t}}[\mathbf{x}\mathbf{x}^\top]$, Eq. (19) can be rewritten via the cyclic property of trace:

$$\Delta_{\text{degrad}} = \operatorname{tr}\big(\Delta \mathbf{W}_i \mathbf{C}^{t-1} \Delta \mathbf{W}_i^\top\big). \tag{20}$$

To balance plasticity and stability, we formulate the learning objective as a soft-margin constrained optimization:

$$\min_{\Delta \mathbf{W}_i} \; \mathcal{L}(\mathbf{W}_i + \Delta \mathbf{W}_i) + \lambda \max\big(0, \; \operatorname{tr}(\Delta \mathbf{W}_i \mathbf{C}^{t-1} \Delta \mathbf{W}_i^\top) - \epsilon\big), \tag{21}$$

where $\epsilon > 0$ defines the tolerance budget for functional deviation and $\lambda > 0$ controls regularization strength.

### C.2. Constrained Formulation and Riemannian Update

By the theory of exact penalty methods, for sufficiently large $\lambda$, the penalized problem in Eq. (21), which employs a hinge penalty on the degradation measure, is equivalent to the following hard-constrained optimization:

$$\begin{aligned} \min_{\Delta \mathbf{W}_i} \quad & \mathcal{L}(\mathbf{W}_i + \Delta \mathbf{W}_i) \\ \text{s.t.} \quad & \operatorname{tr}(\Delta \mathbf{W}_i \mathbf{C}^{t-1} \Delta \mathbf{W}_i^\top) \leq \epsilon. \end{aligned} \tag{22}$$

In stochastic optimization, we approximate the loss via first-order Taylor expansion:

$$\mathcal{L}(\mathbf{W}_i + \Delta \mathbf{W}_i) \approx \mathcal{L}(\mathbf{W}_i) + \langle \nabla_{\mathbf{W}_i} \mathcal{L}, \Delta \mathbf{W}_i \rangle, \tag{23}$$

where $\langle \mathbf{A}, \mathbf{B} \rangle = \operatorname{tr}(\mathbf{A}^\top \mathbf{B})$. Substituting this into the Lagrangian of (22):

$$\mathcal{J} = \langle \nabla_{\mathbf{W}_i} \mathcal{L}, \Delta \mathbf{W}_i \rangle + \lambda \big( \operatorname{tr}(\Delta \mathbf{W}_i \mathbf{C}^{t-1} \Delta \mathbf{W}_i^\top) - \epsilon \big), \tag{24}$$

and minimizing over $\Delta \mathbf{W}_i$ yields the stationarity condition:

$$\nabla_{\mathbf{W}_i} \mathcal{L} + 2\lambda \Delta \mathbf{W}_i \mathbf{C}^{t-1} = \mathbf{0}. \tag{25}$$

Assuming $\mathbf{C}^{t-1}$ is positive definite on its support, we solve for the update direction:

$$\Delta \mathbf{W}_i = -\frac{1}{2\lambda} \nabla_{\mathbf{W}_i} \mathcal{L} \, (\mathbf{C}^{t-1})^{-1}. \tag{26}$$

This coincides with the *Riemannian gradient* on the manifold equipped with metric tensor $\mathbf{G} = \mathbf{C}^{t-1}$:

$$\nabla_{\mathcal{M}} \mathcal{L} = \nabla_{\mathbf{W}_i} \mathcal{L} \, \mathbf{G}^{-1}. \tag{27}$$

Crucially, we couple the dual variable $\lambda$ to the scheduled learning rate $\eta_t$ via:

$$\lambda_t = \frac{1}{2\eta_t}, \tag{28}$$

yielding the practical update rule:

$$\Delta \mathbf{W}_i = -\eta_t \, \nabla_{\mathbf{W}_i} \mathcal{L} \, (\mathbf{C}^{t-1})^{-1}. \tag{29}$$

This design enables *stage-adaptive drift control* without extra hyperparameters:

- Early training: large $\eta_t \Rightarrow$ small $\lambda_t \Rightarrow$ relaxed constraint (promotes plasticity).

- Late training: decaying $\eta_t \Rightarrow$ large $\lambda_t \Rightarrow$ tightened constraint (enhances stability).

The dynamic trade-off aligns with the principle that continual learners should prioritize plasticity when far from convergence and stability near the solution, and it remains fully compatible with standard training pipelines.

### C.3. Implicit Absorption of Soft-margin Threshold

The explicit soft-margin threshold $\epsilon$ does not appear in the update rule (29), yet it is implicitly controlled through the learning rate schedule. From Eq. (26), the degradation magnitude is:

$$\begin{aligned}
\Delta_{\text{degrad}} &= \text{tr}\big(\Delta \mathbf{W}_i \mathbf{C}^{t-1} \Delta \mathbf{W}_i^{\top}\big) \\
&= \frac{1}{4\lambda_t^2} \text{tr}\big(\nabla_{\mathbf{W}_i} \mathcal{L} \, (\mathbf{C}^{t-1})^{-1} \nabla_{\mathbf{W}_i} \mathcal{L}^{\top}\big).
\end{aligned} \tag{30}$$

In the constrained formulation (22), we require $\Delta_{\text{degrad}} \leq \epsilon$. Under the linearized objective (which is tight near $\mathbf{W}_i$), the optimal update saturates this bound, i.e., $\Delta_{\text{degrad}} = \epsilon$. Solving for $\lambda_t$ yields:

$$\lambda_t = \frac{1}{2\sqrt{\epsilon}} \sqrt{\text{tr}\big(\nabla_{\mathbf{W}_i} \mathcal{L} \, (\mathbf{C}^{t-1})^{-1} \nabla_{\mathbf{W}_i} \mathcal{L}^{\top}\big)}. \tag{31}$$

By coupling $\lambda_t = 1/(2\eta_t)$ (Eq. (28)), the effective threshold becomes time-varying:

$$\epsilon_t = \eta_t^2 \cdot \text{tr}\big(\nabla_{\mathbf{W}_i} \mathcal{L} \, (\mathbf{C}^{t-1})^{-1} \nabla_{\mathbf{W}_i} \mathcal{L}^{\top}\big).$$

Thus, standard learning rate decay (e.g., cosine schedule) automatically tightens the drift constraint as training progresses, without requiring manual tuning of $\epsilon$.

## D. Derivation of Feature Sensitivity via AGOP

Let us consider a LoRA expert $i$ with output $f_i(\mathbf{x}) = \Delta \mathbf{W}_i \mathbf{x}$, where $\Delta \mathbf{W}_i \in \mathbb{R}^{d_{\text{out}} \times d_{\text{in}}}$ represents the trainable weight matrix and $\mathbf{x} \in \mathbb{R}^{d_{\text{in}}}$ is the input token. The *Neural Feature Matrix* (NFM) for this expert at layer $\ell$ is defined as:

$$\mathbf{W}_i^{(\ell)\top} \mathbf{W}_i^{(\ell)},$$

where $\mathbf{W}_i^{(\ell)}$ denotes the weight matrix at layer $\ell$.

NFM is proportional to the *Average Gradient Outer Product* (AGOP) matrix:

$$\mathbf{W}_i^{(\ell)\top} \mathbf{W}_i^{(\ell)} \propto \left( \frac{1}{n} \sum_{p=1}^{n} \sum_{j=1}^{m} \nabla_{\mathbf{u}_i^{(\ell)j}} \hat{f}(\mathbf{x}^{(p)}) \nabla_{\mathbf{u}_i^{(\ell)j}} \hat{f}(\mathbf{x}^{(p)})^{\top} \right)^{a},$$

where $\hat{f}$ is the trained network, $\mathbf{u}_i^{(\ell)j}$ denotes the $j$-th input component to layer $\ell$, $n$ is the number of samples, $m$ is the number of input components, and $a > 0$ is an exponent parameter. we set $a = \frac{1}{2}$ for all empirical results.

For our linear LoRA expert, with $\theta_i = \text{vec}(\Delta \mathbf{W}_i)$ denoting the vectorized parameters, the gradient computation simplifies to:

$$\begin{aligned}
\nabla_{\theta_i} f_i(\mathbf{x}) &= \nabla_{\theta_i} (\Delta \mathbf{W}_i \mathbf{x}) \\
&= \mathbf{x} \otimes \mathbf{I}_{d_{\text{out}}},
\end{aligned} \tag{32}$$

*Table 5.* Details of datasets used in TriGap benchmark.

| Dataset | Train | Test | Domain Description |
|---|---|---|---|
| PMCVQA | 40000 | 3000 | Medical image analysis and diagnosis |
| DocVQA | 30000 | 3000 | Document understanding and text extraction |
| ChartQA | 25000 | 3000 | Chart and graph reasoning |
| IconQA | 10000 | 3000 | Icon comprehension |
| InfographicVQA | 20000 | 3000 | Infographic information extraction |
| ArxivQA | 10000 | 3000 | Academic paper figure analysis |
| Roadside | 40000 | 3000 | Autonomous driving scene understanding |
| ChemVQA | 40000 | 3000 | Molecular structure analysis |
| FloodNetVQA | 10000 | 3000 | Disaster scene assessment |
| CLEVR | 10000 | 3000 | Mathematical reasoning on synthetic scenes |

where $\otimes$ denotes the Kronecker product and $\mathbf{I}_{d_{\text{out}}}$ is the identity matrix of dimension $d_{\text{out}}$.

The trace of the AGOP matrix, which measures the total functional sensitivity across all directions, is then:

$$\text{tr}(\text{AGOP}) = \mathbb{E}_{\mathbf{x} \sim \mathcal{D}}[\|\nabla_{\theta_i} f_i(\mathbf{x})\|^2]$$

$$= \mathbb{E}_{\mathbf{x} \sim \mathcal{D}}[\|\mathbf{x} \otimes \mathbf{I}_{d_{\text{out}}}\|^2] \tag{33}$$

$$= \mathbb{E}_{\mathbf{x} \sim \mathcal{D}}[\|\mathbf{x}\|^2 \cdot \|\mathbf{I}_{d_{\text{out}}}\|_F^2] \tag{34}$$

$$= d_{\text{out}} \cdot \mathbb{E}_{\mathbf{x} \sim \mathcal{D}}[\|\mathbf{x}\|^2], \tag{35}$$

where $\| \cdot \|_F$ denotes the Frobenius norm.

This derivation shows that for linear LoRA experts, the trace of AGOP is proportional to the expectation of the squared norm of the input. Therefore, we can estimate feature sensitivity through a running average of $\|\mathbf{x}\|^2$ rather than explicitly computing high-dimensional gradients.

# E. Details of the TriGap

TriGap is designed to provide a more comprehensive and challenging evaluation suite for Multimodal Continual Instruction Tuning (MCIT) by systematically addressing three critical dimensions that are often overlooked in existing benchmarks:

**(1) Extended Sequence & Instruction Gap.** TriGap comprises 10 sequential tasks. This longer horizon amplifies router drift and expert drift, providing a stricter test of long-term stability. Instruction formats also vary widely, including multiple-choice answers (PMCVQA, ArxivQA), numerical values (DocVQA, CLEVR), single-word or short-phrase responses (ChartQA, IconQA, Roadside, FloodNetVQA), chemical formulas (ChemVQA), and capitalized phrases (InfographicVQA). Such diversity challenges the model to preserve task-specific conventions while adapting to new styles.

**(2) Task Span & Difficulty Gap.** The benchmark spans diverse vision-language capabilities, from chart understanding and icon comprehension to scientific reasoning and real-world scene analysis (see Fig. 9).

**(3) Dataset Scale Gap.** As summarized in Tab. 5, per-task training sizes vary from 10k to 40k samples, simulating realistic non-uniform data streams. Large-scale tasks exert stronger optimization pressure that may overwrite prior knowledge, while small-scale tasks offer limited adaptation signals.

## 1. PMCVQA

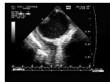 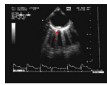
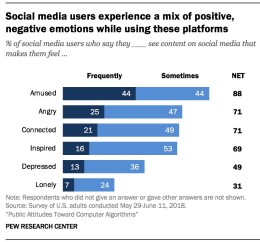

**Question:** What is the feature of the aortic lesion in the left lower panel? Answer with the option's letter.

**Choices:**

A.    Multiple plaque

B.    Mobile part

C.    Large plaque

D.    Small plaque

**Ground-truth:** C

## 2. DocVQA

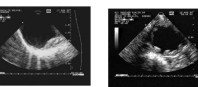

**Question:** What is written in the Zip code Field ?

**Ground-truth:** 91301

## 3. ChartQA

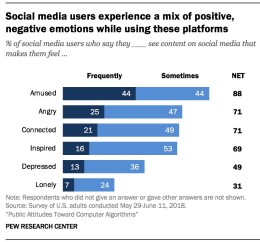

**Question:** What was the 4th most popular emotion? Answer the question using a single word or phrase.

**Ground-truth:** Inspired

## 4. IconQA

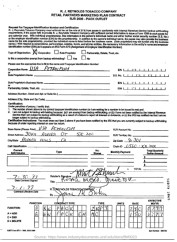

**Question:** How many shapes are green?

**Ground-truth:** 5

## 5. InfographicVQA

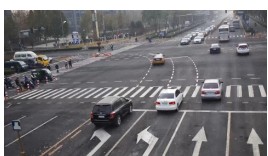

**Question:** Which brain system is taking care of behaviours like conscious, slow , logical?

**Ground-truth:** SYSTEM 2' THINKING

## 6. ArxivQA

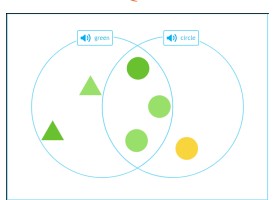

**Question:** What trend can be observed in the vortex density as the temperature range shifts from cold to hot in graphs (b) and (d)?

**Choices:**

A.    Vortex density increases with the hot temperature range.

B.    Vortex density decreases with the hot temperature range.

C.    Vortex density remains constant regardless of the temperature range.

D.    Vortex density shows a random pattern with no clear trend related to the temperature range.

**Ground-truth:** A

## 7. Roadside

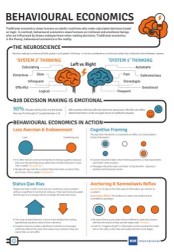

**Question:** Which vehicles are stopped at the closest crosswalk? Answer the question using a single word or phrase.

**Ground-truth:** A silver car and a white car.

## 8. ChemVQA

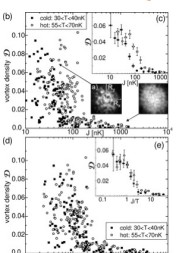

**Question:** What is the molecular formula of this molecule?

**Ground-truth:** C9H25NO8S2

## 9. FloodNetVQA

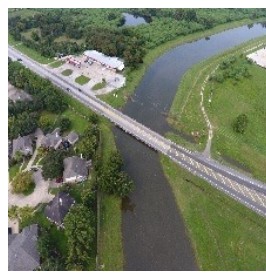

**Question:** Do the rescuers need to provide help urgently?

**Ground-truth:** no

## 10. CLEVR

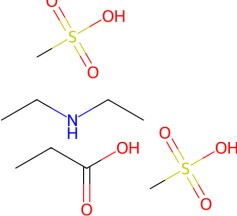

**Question:** Subtract all purple cylinders. Subtract all blue matte blocks. How many objects are left?

**Ground-truth:** 7

*Figure 9.* Representative examples from the 10 datasets included in the TriGap benchmark.

*Table 6.* Hyperparameter sensitivity on UCIT. All values are average accuracy (%). SAME shows robustness to moderate variations across all three key hyperparameters.

| Parameter | Values Tested | Selected | Results |
|---|---|---|---|
| $\delta$ | $\{0.80, 0.85, 0.90, 0.95\}$ | 0.90 | 66.5 / 66.4 / **67.4** / 66.9 |
| $\mu$ | $\{0.09, 0.9, 9\}$ | 0.9 | 62.3 / **67.4** / 61.4 |
| $\tau_{\text{score}}$ | $\{0.075, 0.1, 0.125\}$ | 0.1 | 65.7 / **67.4** / 66.3 |

*Table 7.* Per-task accuracy under Order 2: SciQA $\rightarrow$ Image $\rightarrow$ GQA $\rightarrow$ OCRVQA $\rightarrow$ REC $\rightarrow$ VQAv2 $\rightarrow$ VizWiz $\rightarrow$ TextVQA.

| Method | SciQA | Image | GQA | OCRVQA | REC | VQAv2 | VizWiz | TextVQA | Avg |
|---|---|---|---|---|---|---|---|---|---|
| MoELoRA | 0 | 36.1 | 55.8 | 55.9 | 63.1 | 62.8 | 45.5 | 58.5 | 47.2 |
| SAME | **78.1** | **89.7** | **60.8** | **63.4** | **60.1** | **66.1** | **57.2** | **61.1** | **67.0** |

*Table 8.* Per-task accuracy under Order 3: OCRVQA $\rightarrow$ VQAv2 $\rightarrow$ REC $\rightarrow$ VizWiz $\rightarrow$ GQA $\rightarrow$ Image $\rightarrow$ TextVQA $\rightarrow$ SciQA.

| Method | OCRVQA | VQAv2 | REC | VizWiz | GQA | Image | TextVQA | SciQA | Avg |
|---|---|---|---|---|---|---|---|---|---|
| MoELoRA | 29.9 | 62.2 | 59.1 | 47.4 | 56.3 | 45.3 | 56.9 | 77.7 | 54.3 |
| SAME | **62.8** | **65.7** | **58.6** | **53.9** | **61.3** | **90.1** | **62.4** | **80.1** | **66.8** |

# F. Hyperparameter Sensitivity and Selection

As shown in Tab. 6, SAME is robust to moderate hyperparameter variations:

- $\delta$: Accuracy varies by only 1.0% across $\delta \in [0.80, 0.95]$, with optimal performance at $\delta = 0.90$. This stability stems from our sliding-window scaling: when singular values are clustered, the exact placement of $\delta$ has minimal impact because updates for such directions are scaled by factors close to 1.

- $\mu$: The optimal value $\mu = 0.9$ balances numerical stability and effective preconditioning. Extremely small $\mu = 0.09$ amplifies noise in near-singular directions, while large $\mu = 9$ overly suppresses updates.

- $\tau_{\text{score}}$: The selected $\tau_{\text{score}} = 0.1$ provides the best trade-off. Freezing too few experts leaves more parameters vulnerable to drift, while freezing too many limits plasticity for new tasks.

All hyperparameters are fixed across tasks and benchmarks after selection via 20% random sampling of UCIT training data, ensuring fair comparison without per-benchmark tuning.

# G. Robustness to Task Ordering

As shown in Tab. 7 and Tab. 8, SAME achieves stable performance across all three orderings, with a maximum deviation of only 0.2%. In contrast, MoELoRA varies by 7.1% across orderings, indicating strong dependence on task sequence.

Notably, under Order 2, MoELoRA suffers complete forgetting on ScienceQA (0% accuracy), where error analysis reveals that 72.5% of predictions are semantically correct but incorrectly formatted in lowercase (e.g., "july" instead of uppercase "A"). This suggests that baseline methods not only forget task knowledge but also lose instruction-following capabilities under unfavorable orderings. SAME mitigates this via spectral-aware routing and adaptive expert activation, which jointly preserve both semantic competence and task-specific output conventions.

*Table 9.* Component-wise and design choice ablations on UCIT. Each row isolates a single modification while keeping other components fixed.

| Variant | Avg Acc (%) |
|---|---|
| **Design choices** | |
| Global mean scaling | 63.3 |
| Sliding-window scaling (ours) | **67.1** |
| Fixed-ratio (50%) expert freezing | 60.4 |
| Adaptive freezing (ours) | **67.1** |
| **Projection-based comparisons** | |
| SAME (anisotropic scaling) | **67.12** |
| w/ hard projection (GPM-style) | 61.72 |
| w/ constant subspace scaling (VPT-CPG-style) | 61.83 |

# H. Ablation Studies and Design Choices

Tab. 9 validates each design choice with quantitative evidence:

**Scaling strategy.** Sliding-window scaling outperforms global mean by $+3.8\%$. The core difference is how singular values are aggregated for scaling:

$$\text{Global: } \alpha_i = \frac{1}{\bar{\sigma}}, \ \bar{\sigma} = \frac{1}{d}\sum_{j=1}^{d}\sigma_j \quad \text{vs.} \quad \text{Ours: } \alpha_i = \frac{1}{\hat{\sigma}_i}, \ \hat{\sigma}_i = \frac{1}{k}\sum_{j=i-k+1}^{i}\sigma_j \tag{36}$$

Our sliding-window estimate $\hat{\sigma}_i$ captures local spectral density, enabling finer-grained control: directions within clustered singular values receive scaling factors close to 1, reducing sensitivity to the exact energy threshold.

**Freezing score formulation.** The subtraction-based score is better. The two formulations differ as follows:

$$\text{Ratio: } S(i) = \frac{\tilde{\mathcal{U}}(i)}{\tilde{\mathcal{F}}^{\text{pre}}(i) + \epsilon} \quad \text{vs.} \quad \text{Ours: } S(i) = \tilde{\mathcal{U}}(i) - \tilde{\mathcal{F}}^{\text{pre}}(i) \tag{37}$$

Ratios become unstable when $\tilde{\mathcal{F}}^{\text{pre}}(i) \approx 0$ (common in early tasks), requiring careful tuning of $\epsilon$. Subtraction remains numerically robust and directly encodes the intuition: freeze experts with low current utility but high historical importance.

**Expert freezing strategy.** Adaptive freezing outperforms fixed-ratio by $+6.7\%$. The selection criteria are:

$$\text{Fixed: } \mathbb{I}[\text{rank}(i) > 0.5N] \quad \text{vs.} \quad \text{Ours: } \mathbb{I}[S(i) < \tau_{\text{score}}] \tag{38}$$

Fixed-ratio freezing uses a static threshold on expert rank, which may inadvertently freeze critical experts or retain redundant ones. Our adaptive criterion dynamically balances knowledge preservation and plasticity by considering both current-task utilization $\tilde{\mathcal{U}}(i)$ and historical importance $\tilde{\mathcal{F}}^{\text{pre}}(i)$.

**Projection-based comparisons.** Replacing anisotropic scaling with GPM-style (Saha et al., 2021) hard projection or VPT-CPG-style (Lu et al., 2025) constant scaling reduces accuracy by $+5.4\%$ and $+5.29\%$, respectively. This confirms that adaptive anisotropic scaling, not orthogonal projection itself, is key to balancing stability and plasticity in deep MLLMs.

*Table 10.* Normalized routing entropy on CoIN (Layer-15). Lower values indicate more concentrated routing. Enabling adaptive freezing leads to structured specialization.

| Task | 1 | 2 | 3 | 4 | 5 | 6 | 7 | 8 |
|------|------|------|------|------|------|------|------|------|
| w/ Freezing | 0.905 | 0.803 | 0.857 | 0.860 | 0.792 | 0.668 | 0.639 | 0.478 |
| w/o Freezing | 0.934 | 0.914 | 0.925 | 0.906 | 0.905 | 0.905 | 0.941 | 0.956 |

*Table 11.* Router drift causality: performance on Task-1 test set using routers from later checkpoints (experts and representations frozen). Even with fixed experts, router drift alone induces forgetting.

| Router from | T2 | T3 | T4 | T5 | T6 | T7 | T8 |
|-------------|------|------|------|------|------|------|------|
| Routing overlap | 0.933 | 0.929 | 0.926 | 0.931 | 0.934 | 0.912 | 0.927 |
| Acc. drop (%) | 0.8 | 1.1 | 1.4 | 1.6 | 1.5 | 2.1 | 1.3 |

*Table 12.* Forgetting analysis on CoIN. Per-task forgetting $F_s = A_{s,s} - A_{T,s}$ and average forgetting $\bar{F} = \frac{1}{T}\sum_s F_s$. Lower (less negative) values indicate better retention.

| Method | SciQA | TextVQA | ImageNet | GQA | VizWiz | REC | VQAv2 | Avg $\bar{F}$ |
|--------|-------|---------|----------|-----|--------|-----|-------|---------------|
| MoELoRA | -11.91 | -12.85 | -60.79 | -14.41 | -15.30 | -18.01 | -9.62 | -19.04 |
| HiDe-LLaVA | **-1.43** | -8.95 | -13.42 | -6.34 | -6.29 | **-5.16** | **-3.12** | -6.38 |
| SAME | -2.01 | **-4.69** | **-2.21** | **-5.89** | **-4.25** | -5.71 | -4.83 | **-4.23** |

## I. Expert Dynamics and Specialization Analysis

Tab. 10 shows that enabling adaptive freezing leads to a progressive decrease in routing entropy, with the reduction magnitude increasing for later tasks. This trend reflects our design: as training progresses, experts that accumulate important knowledge from earlier tasks are temporarily frozen to protect that knowledge. Consequently, routing becomes more concentrated on the remaining active experts.

This pattern represents *structured specialization* rather than random siloing: experts dynamically adjust their participation based on task relevance and historical importance. For instance, by Task 8, entropy drops to $0.478$ with freezing enabled, compared to $0.956$ without freezing, indicating focused expert utilization without sacrificing cross-task transfer.

Tab. 11 isolates the causal effect of router drift: even with Task-1 representations and experts frozen, routing overlap gradually decreases ($0.933 \rightarrow 0.927$) while accuracy drops by up to $2.1\%$ (at T7). This confirms that router weight drift alone can induce forgetting, independent of backbone or expert representation shifts. SAME mitigates this via spectral-aware routing, which constrains router updates to task-relevant subspaces.

## J. Quantitative Forgetting Analysis

Tab. 12 quantifies retention beyond final accuracy. SAME achieves the lowest average forgetting ($\bar{F} = -4.23\%$), substantially outperforming MoELoRA ($-19.04\%$) and HiDe-LLaVA ($-6.38\%$). Notably, SAME exhibits particularly strong retention on visually intensive tasks such as ImageNet ($-2.21\%$ vs. $-60.79\%$ for MoELoRA), indicating that curvature-aware scaling effectively protects experts from destructive updates along historically important directions. The consistently smaller forgetting across all tasks further confirms that spectral-aware routing and adaptive expert activation jointly mitigate both router drift and expert drift, enabling more stable knowledge preservation throughout the continual tuning process.

*Table 13.* OOD evaluation: models trained on UCIT, evaluated on MMMU test sets. Higher accuracy indicates better cross-benchmark generalization.

| Method | Art&Design | Business | Science | Health&Med | Humanities | Tech&Eng | Avg |
|---|---|---|---|---|---|---|---|
| Zero-Shot | 43.85 | 25.35 | 23.74 | 32.93 | **48.47** | 28.16 | 33.75 |
| MoELoRA | 44.37 | 26.39 | 24.98 | 35.28 | 47.69 | 30.32 | 34.83 |
| SAME | **47.98** | **26.51** | **25.69** | **36.51** | 47.46 | **33.96** | **36.35** |

*Table 14.* Efficiency comparison: MoELoRA vs SAME.

| Inference speed | Baseline | SAME |
|---|---|---|
| Throughput | 1.03 it/s | **1.26 it/s** |

## K. Cross-Benchmark OOD Generalization

Tab. 13 demonstrates transferable representations learned by SAME:

- **Overall gain**: SAME achieves the highest average accuracy (36.35%), outperforming MoELoRA by +1.52% and zero-shot baseline by +2.60%.

- **Domain-specific improvements**: Gains are consistent across domains, with particularly strong improvements in Art&Design (+4.13% over zero-shot) and Tech&Engineering (+3.64%). These domains require fine-grained visual reasoning and structured output formatting, capabilities that SAME effectively preserves through curvature-aware scaling and adaptive expert activation.

- **Interpretation**: These results suggest that stabilizing expert selection and protecting historical knowledge helps SAME retain generalizable capabilities that transfer to unseen task distributions, rather than overfitting to the training benchmark.

## L. Efficiency and Computational Overhead

Tab. 14 quantifies the practical benefits of SAME:

**Memory overhead.** Using randomized low-rank decomposition (Halko et al., 2011), we store only top-64 singular directions per layer, reducing covariance storage from 32 MB to ∼0.5 MB per layer (64× compression). The full SAME state fits within a 139 MB checkpoint, with only a 0.68% accuracy drop compared to full-rank storage.

**Inference speed.** Stabilized routing enables reliable top-$k$ expert selection at inference time. In contrast, unstable baselines like MoELoRA must avoid top-$k$ routing due to drift-induced misassignment, falling back to dense evaluation over all experts. SAME achieves 22% faster inference (1.03→1.26 it/s) while maintaining higher accuracy, demonstrating that stabilization improves both performance and efficiency.

## M. Further Analysis of Router Drift

We examine how expert utilization varies across network depth to understand layer-specific routing dynamics. As shown in Figure 10, routing behavior changes systematically from shallow to deep layers.

In shallow layers (e.g., Layer 0), expert utilization remains nearly uniform across all tasks. This indicates that early layers act as general-purpose routers, distributing computation evenly without strong task specialization.

In contrast, deeper layers develop pronounced task-specific preferences. At Layer 10, Task 1 heavily favors Expert 7 (weight 0.225) and Expert 5 (0.165), while Task 7 shows a more balanced but still skewed pattern. By Layer 20, specialization becomes even stronger: Expert 3 dominates on Task 1 (0.148), Expert 6 on Tasks 3 and 5 (0.180 and 0.196), and Expert 6 becomes the clear favorite on Task 7 (0.242). These patterns confirm that routing decisions become increasingly task-dependent with depth.

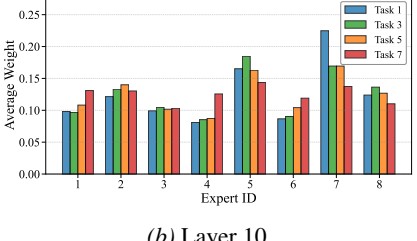
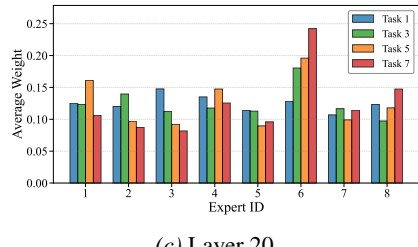

*(a) Layer 0*      *(b) Layer 10*      *(c) Layer 20*

*Figure 10.* Layer-wise expert utilization patterns.

Importantly, the same expert plays different roles across layers. For Task 1, Expert 3 receives only 0.124 weight in Layer 0 but rises to 0.148 in Layer 20, becoming the most activated expert. Conversely, Expert 7 dominates in Layer 10 (0.225) but drops to 0.107 in Layer 20. This layer-wise variation indicates that router drift is weaker in early layers and grows stronger in deeper ones, where routing becomes highly task-specific.

These findings have direct implications for stabilization design. Shallow layers are inherently stable due to their uniform routing and require minimal protection. Deep layers, however, demand stronger safeguards against distribution shifts because their specialized routing is easily disrupted. A uniform regularization strategy would therefore be suboptimal: it would unnecessarily constrain shallow layers (hurting plasticity) while failing to adequately stabilize deep layers (allowing drift). Our spectral-aware routing addresses this by adapting gradient projections per layer, which preserves history-critical directions where needed while allowing flexible adaptation in task-relevant subspaces.

## N. Comprehensive Study of MCIT methods

In this section, we provide details of the methods compared in the main paper. The specifics of each compared method are outlined as follows:

- **Replay-LoRA**: A rehearsal-based baseline that integrates a fixed-size episodic memory buffer with LoRA fine-tuning. During each incremental task, it stores a representative subset of historical multimodal instructions and jointly optimizes the low-rank adapters using both current and replayed samples. This explicit data rehearsal strategy effectively constrains parameter drift and mitigates catastrophic forgetting while preserving parameter efficiency.

- **MoELoRA**: This method extends LoRA-based fine-tuning to a Mixture-of-Experts architecture for continual instruction tuning, where each task activates a subset of LoRA experts via a learnable router, enabling parameter-efficient adaptation across sequential tasks without rehearsal.

- **Continual LLaVA**: This approach introduces a low-rank pool of proxy-increment embedding pairs to support rehearsal-free continual instruction tuning. For each input instruction, it selects relevant embeddings based on textual similarity and aggregates previously selected embeddings via learnable weights, enabling efficient knowledge integration across tasks.

- **HiDe-LLaVA**: Based on CKA similarity analysis revealing distinct representation patterns between top and lower transformer layers, this method hierarchically decouples model adaptation: the top layer undergoes task-specific LoRA expansion with dual-modality anchor matching for expert selection, while lower layers fuse LoRAs across tasks to preserve general knowledge without router training.

- **ModalPrompt**: A prompt-based framework that constructs task-specific prompts and leverages dual-modality guidance for two purposes: prompt fusion during training to transfer knowledge from semantically similar tasks, and prompt selection during inference to control computational complexity. The method maintains inference efficiency by selecting only $k$ relevant prompts from a shared pool regardless of task count.

- **SEFE**: This method addresses forgetting via answer style diversification and RegLoRA, which applies regularization to top-$M\%$ elements of LoRA weight update matrices to preserve critical historical knowledge.

- **LLaVA-CMoE**: A continual MoE framework featuring probe-guided knowledge extension that dynamically allocates experts only where capacity gaps exist by monitoring probe activation frequencies, and a probabilistic task locator that uses VAE-based reconstruction probability to select task-specific routers without explicit task-ID during inference.

- **ProgLoRA**: This method introduces a progressive LoRA pool for multimodal continual instruction tuning, where a new LoRA block is trained and added for each incremental task while all previously learned LoRA blocks are frozen to preserve acquired knowledge. To effectively leverage knowledge from historical tasks, ProgLoRA employs task-aware allocation that selects and fuses relevant LoRA blocks based on task similarity. Additionally, it incorporates task recall to constrain model updates and further mitigate forgetting on prior tasks. Two variants are provided: ProgLoRA (static) for idealized settings with known task identity during inference, and ProgLoRA (dynamic) for realistic settings without task identity.

- **CL-MoE**: A dual momentum Mixture-of-Experts framework designed for continual MLLM adaptation. It introduces a Dual-Router MoE (RMoE) that jointly employs task-level and instance-level routers to robustly allocate global and local experts. Coupled with a Dynamic Momentum MoE (MMoE), it categorizes experts into task-shared and task-specific types and dynamically updates their parameters via a momentum-based interpolation between historical and current knowledge. This mechanism effectively alleviates catastrophic forgetting and enhances both forward and backward transfer without relying on replay buffers.

