# OpenReview forum: "SAME: Stabilized Mixture-of-Experts for Multimodal Continual Instruction Tuning"
_ICML.cc/2026/Conference — ICML 2026 regular_

### Official Review · Reviewer_sqZx · 2026-03-11

**Soundness:** 3
**Presentation:** 3
**Significance:** 3
**Originality:** 3
**Overall Recommendation:** 4
**Confidence:** 2

**Summary:**

This paper studies multimodal continual instruction tuning for MLLMs and argues that MoE based continual tuning suffers from **router drift** and **expert drift**. The proposed method **SAME** targets these with: 1) *spectral aware routing* that limits router updates using a low-rank covariance subspace decomposition, 2) *curvature aware scaling* that preconditions expert updates using inverse historical input covariance, and 3)*adaptive expert activation/freezing* that freezes experts that are low utility for the current task but historically important. Experiments on the CoIN benchmark report improved the final average over the MCIT baselines.

**Compliance With Llm Reviewing Policy:**

Affirmed.

**Final Justification:**

The paper is well-motivated, with clear separation of router drift and expert drift and strong CoIN results. My two concerns: top-k sensitivity and missing forgetting metrics, were both addressed in the rebuttal. The routing overlap analysis and per-task forgetting results strengthen the empirical story considerably. However, the theoretical justification for spectral-aware routing under discrete top-k selection remains partially incomplete, and I maintain my score of 4 (Weak Accept). The rebuttal reinforced rather than changed my prior assessment.

**Key Questions For Authors:**

NA

**Limitations:**

yes

**Strengths And Weaknesses:**

**Strengths**:
The paper provides a clear separation into router drift vs expert drift, which provides useful insights for analyzing continual MoE behavior. Spectral aware routing aims to stabilize routing behavior, whereas curvature aware scaling seeks to reduce destructive expert updates, and adaptive activation/freezing reduces unnecessary interference and improves efficiency. The empirical results on CoIN are strong. The rerouting protocol used to separate router drift from expert drift is meaningful.

**Weakness**:
The theoretical justification is partly convincing. The subspace projection argument in spectral aware routing is plausible, but top-k expert selection can flip with small logit changes. The evaluation emphasises final accuracy, but continual learning performance would be better characterized with explicit forgetting metrics.

---

> ### Author Rebuttal · Authors · 2026-03-31
>
> ## W1: top-k expert selection
>
> Thank you for this insightful comment. We acknowledge that top-k selection is inherently sensitive to logit perturbations. However, SAME stabilizes routing through the joint effect of Spectral-aware Routing and Adaptive Expert Activation:
>
> 1.  Router Bias & Expert Specialization: Router logits for a given task are not uniformly distributed; we observe consistent routing preferences where certain experts are repeatedly selected for task-specific inputs. Since our Adaptive Expert Activation tracks utilization during training, these frequently activated experts absorb the majority of task-specific knowledge, establishing expert-task specialization.
>
> 2.  Router Drift Control: Spectral-aware Routing ensures that routing preferences for prior tasks remain stable during new task training.This constraint directly reduces top-k flipping, preserving expert specialization and mitigating router drift.
>
> Together, these mechanisms ensure that at inference time, task samples are consistently routed to the same experts that were active during their training, preserving expert specialization and mitigating forgetting.
>
> Empirically, we measure routing overlap (the consistency of expert selection on Task 1 inputs after learning subsequent tasks). SAME maintains substantially higher overlap than MoELoRA across the task sequence:
>
> | Method   | Task 2   | Task 4   | Task 6   | Task 8   |
> | -------- | -------- | -------- | -------- | -------- |
> | MoELoRA  | 86.3     | 83.4     | 81.4     | 77.9     |
> | **SAME** | **91.1** | **87.0** | **84.7** | **81.2** |
>
> ## W2: final accuracy vs. explicit forgetting metrics
>
> Thank you for this valuable suggestion. We agree that explicit forgetting metrics provide a more comprehensive characterization of continual learning performance beyond final average accuracy. Following standard continual-learning evaluation, we additionally report both per-task forgetting and average forgetting on CoIN to better quantify how well each method preserves previously learned knowledge over the full task sequence.
>
> Per-Task Forgetting ($F_s$): It measures the performance change on task $s$ after completing all $T$ tasks [1], defined as $F_s = A_{s,T} - A_{s,s}$, where $A_{s,s}$ and $A_{s,T}$ denote the accuracy on task $s$ immediately after training on it and after the final task, respectively. $F_s < 0$ indicates forgetting (performance drop). Higher $F_s$ reflects better anti-forgetting capability.
>
> Average Forgetting ($\bar{F}$): It summarizes overall retention across the task sequence by averaging per-task forgetting over all $T-1$ prior tasks: $\bar{F} = \frac{1}{T-1} \sum_{s=1}^{T-1} F_s$. A higher (less negative) $\bar{F}$ indicates stronger global stability and less catastrophic forgetting.
>
> Below we report the forgetting results on the CoIN benchmark:
>
> | Method     | ScienceQA | TextVQA   | ImageNet  | GQA       | VizWiz    | REC       | VQAv2     | Avg. Forgetting |
> | ---------- | --------- | --------- | --------- | --------- | --------- | --------- | --------- | --------------- |
> | MoELoRA    | -11.91    | -12.85    | -60.79    | -14.41    | -15.30    | -18.01    | -9.62     | -19.04          |
> | HiDe-LLaVA | **-1.43** | -8.95     | -13.42    | -6.34     | -6.29     | **-5.16** | **-3.12** | -6.38           |
> | **SAME**   | -2.01     | **-4.69** | **-2.21** | **-5.89** | **-4.25** | -5.71     | -4.83     | **-4.23**       |
>
> These results show that SAME achieves the best overall retention, with the highest (least negative) average forgetting score, improving over MoELoRA by 14.81% and over HiDe-LLaVA by 2.15%. The gain is especially pronounced on ImageNet, where SAME reduces forgetting from -60.79 to -2.21, indicating that our method substantially alleviates severe long-horizon degradation on visually distinctive tasks.
>
> Across tasks, SAME also delivers consistently stronger retention on ScienceQA, ImageNet and VizWiz, and remains competitive on the remaining tasks. Although HiDe-LLaVA is slightly better on REC and VQAv2, SAME still obtains the best overall average forgetting, showing that its improvements are not confined to a single task type but reflect a stronger balance between adaptation and retention throughout the sequence.
>
> Overall, these forgetting results are consistent with our main claim: SAME improves not only the final average accuracy, but also the underlying learning dynamics by better preserving previously acquired knowledge over time.
>
>
>
> [1] A Comprehensive Survey of Continual Learning: Theory, Method and Application. arXiv:2302.00487, 2023.

---

> > ### Author Rebuttal · Reviewer_sqZx · 2026-03-31
> >
> > Thank you for your response. My main concerns were the sensitivity of top-k expert selection and the absence of explicit forgetting metrics, which were addressed in the rebuttal. However, I am keeping my overall assessment and score unchanged.

---

> > > ### Author Response · Authors · 2026-04-01
> > >
> > > Thank you for your kind response. We are glad to hear that our clarifications addressed your concerns. We sincerely appreciate the time and effort you devoted to reviewing our work.

---

### Official Review · Reviewer_QtnR · 2026-03-12

**Soundness:** 2
**Presentation:** 3
**Significance:** 2
**Originality:** 2
**Overall Recommendation:** 3
**Confidence:** 4

**Summary:**

The paper proposes SAME (Stabilized Mixture-of-Experts) to address catastrophic forgetting in Multimodal Continual Instruction Tuning (MCIT). The authors identify two primary sources of forgetting in MoE architectures: "router drift" (inconsistent expert selection over time) and "expert drift" (overwriting of shared experts). To tackle these, SAME introduces spectral-aware routing to project router updates into orthogonal task-relevant subspaces, curvature-aware scaling (using Riemannian preconditioning) to regularize expert updates, and an adaptive expert activation mechanism to dynamically freeze historically important experts.

**Compliance With Llm Reviewing Policy:**

Affirmed.

**Final Justification:**

I believe this paper may have fairness issues, as there are obvious contradictions in the authors' rebuttals. The authors claim that SAME fully adheres to the CoIN protocol based on LLaVA-1.5-base. However, the comparison results of ModalPrompt and Hide-LLaVA appear to be directly taken from their respective original papers/codebases, both of which use LLaVA-1.5-Instruct. In other words, the rebuttals simultaneously cite both the CoIN-style baseline setup and the baseline data obtained under the instruction setup. This raises a fundamental fairness question: are different methods being compared under different frameworks, yet labeled as a "fair and consistent comparison"? If so, this directly contradicts the authors' claims and undermines the validity of the experimental comparisons. The authors should clearly state the specific LLaVA-1.5 variant used for each method, and whether all baselines are based on the same framework and protocol.

**Key Questions For Authors:**

Question 1: Novelty of the Orthogonal Subspace Projection
While the paper stabilizes the MoE router by decomposing routing dynamics into orthogonal subspaces, relying on orthogonal or null-space projections to mitigate catastrophic forgetting is a heavily explored paradigm in continual learning, as seen in methods like Gradient Projection Memory (GPM) [1]. More critically, a recent concurrent work (VPT-CPG) [2] has already successfully applied exact null-space orthogonal projections to both MoE routers and experts to guarantee stability. Given this, could the authors explicitly clarify their fundamental mechanistic novelty—such as whether the performance gains stem primarily from swapping hard projections for Riemannian scaling or the newly introduced AGOP-based expert freezing—and provide an ablation study that strictly isolates this contribution against a strong orthogonal-subspace baseline?

[1] "Gradient Projection Memory for Continual Learning." ICLR 2021.
[2] "Training Consistent Mixture-of-Experts-Based Prompt Generator for Continual Learning." AAAI 2025.


Question 2: Critical Issue with Data Contamination
There appears to be a severe data contamination issue that undermines the experimental claims, as the paper evaluates the LLaVA-v1.5-7B backbone on the CoIN benchmark (which includes TextVQA, GQA, VQAv2, and OCR-VQA), despite the official LLaVA-1.5 training recipe explicitly including the training splits of these exact datasets in its 665K visual instruction-tuning mixture. Because the model has already observed these images and annotations before the continual learning phase begins, it is impossible to determine whether the reported gains are genuinely due to the SAME architecture or simply the model recalling its pre-trained knowledge. To validate the method's true effectiveness, the authors must quantify this dataset overlap and evaluate SAME on a strictly "clean" initialization (e.g., a pre-alignment checkpoint or a base Vicuna model) that has never been pre-exposed to these downstream tasks.

**Limitations:**

The authors briefly discuss standard limitations regarding ambiguous task boundaries in the conclusion. However, they completely overlook the critical limitation of pre-training data exposure/contamination (as detailed in Question 2). This must be acknowledged and addressed.

**Strengths And Weaknesses:**

Strengths:
1.Sensible Structural Design: Transitioning from traditional rigid null-space projections to a curvature-aware scaling formulation represents a structurally reasonable attempt to provide a softer constraint for expert updates, aiming to balance plasticity and stability.
2.Consideration of Training Overhead: The inclusion of the AGOP-based adaptive expert activation demonstrates a practical awareness of the computational bottlenecks in sequential training for large models. Selectively freezing experts is a logical architectural choice to reduce training costs.

Weaknesses:
1.The submission currently suffers from critical flaws regarding its experimental integrity (data contamination) and the contextualization of its core novelty against highly related concurrent work. Please see the detailed elaborations in the "Key Questions For Authors" section below.

---

> ### Author Rebuttal · Authors · 2026-03-31
>
> ## W1 & Q2: Data contamination
>
> Thank you for this important concern regarding data contamination. We appreciate the opportunity to clarify the validity of our experimental claims.
>
> 1.CoIN is a widely-adopted benchmark in MCIT research. All methods in our comparison share the identical LLaVA-v1.5-7B initialization. Any pre-exposure to CoIN tasks affects all methods equally. Moreover, SAME outperforms the direct zero-shot by 12.61% average accuracy on CoIN, confirming that our architecture contributes meaningful gains beyond pre-trained knowledge recall.
>
> | Method   | Avg   |
> | - | - |
> | Zeroshot | 54.21 |
> | SAME     | 66.82 |
>
> 2.Validation on a strictly clean benchmark (UCIT). To directly address reviewer's contamination concerns, we evaluate SAME on the UCIT benchmark [1], which contains tasks absent from LLaVA's instruction-tuning data.
>
> | Method     | Image-R | ArxivQA | Viz-cap | IconQA | CLEVR | Flickr30k | Avg   |
> | - | - | - | -| - | - | - | - |
> | Zeroshot   | 16.27   | 53.73   | 38.39   | 19.20  | 20.63 | 41.88     | 31.68 |
> | MoELoRA    | 49.87   | 77.63   | 43.65   | 46.40  | 36.47 | 58.34     | 52.06 |
> | HiDe-LLaVA | 80.50    | 89.83   | 48.78   | 62.90  | 47.97 | 55.15     | 64.19 |
> | SAME       | 83.83   | 91.40   | 51.33   | 65.27  | 53.50 | 57.43     | 67.12 |
>
> SAME consistently outperforms baselines across all UCIT tasks, demonstrating that our gains stem from the architecture itself rather than pre-training recall.
>
> ## W1 & Q1 & Limitations: Novelty beyond orthogonal projection
>
> Thank you for this question. We would like to clarify that orthogonal projection is merely a common optimization tool; the fundamental novelty lies in how gradients are processed within the projected subspace. This design choice is central to SAME's superiority over existing projection-based methods in balancing stability and plasticity.
>
> Most existing methods employ either hard projection (discarding gradients in the old subspace entirely) or constant subspace scaling (multiplying the entire projected subspace by a single global factor $\eta$). While effective in lightweight settings, these strategies face critical limitations in large-scale MLLMs:
>
> 1.  Plasticity Bottleneck: Hard projection severely restricts the model's ability to learn complex new tasks by zeroing out some update directions.
> 2.  Prohibitive Tuning Cost: Constant scaling requires careful tuning of hyperparameters (e.g., null-space thresholds and $\eta$). Constant scaling methods like VPT-CPG are designed to control router and expert drift within a single MoE module, where $\eta$ can be tuned once to balance stability and plasticity. However, this design does not scale to MLLMs, which contain dozens of MoE layers with heterogeneous drift dynamics. A uniform $\eta$ cannot accommodate layer-wise variations, while tuning $\eta$ per layer becomes prohibitively expensive and impractical. More critically, representational drift accumulates across layers: even minor instability in early layers propagates and amplifies downstream, making isotropic scaling fundamentally insufficient for deep architectures.
>
> In contrast, SAME employs adaptive anisotropic scaling across both Spectral-aware Routing  and Curvature-aware Scaling, fundamentally challenging the isotropic assumption inherent in prior works. While VPT-CPG applies a uniform relaxation factor across the entire subspace, this contradicts the objective reality of deep networks, where representational drift is inherently direction-dependent. SAME recognizes this anisotropy, scaling each singular direction individually via sliding windows (router) and historical curvature (experts). This enables fine-grained control, suppressing unstable high-variance directions while preserving updates in stable ones.
>
> Furthermore, the sliding window captures local spectral density, enhancing robustness. When some singular values are clustered, the exact placement of the energy threshold within this cluster has minimal impact, as the update for such directions is scaled by a factor close to 1, rendering the suppression effect negligible.
>
> Below we present the results on the UCIT after replacing our scaling with GPM-style hard projection and VPT-CPG-style constant subspace scaling, respectively, while strictly adopting their recommended hyperparameter settings for a fair comparison.
>
> | method | Image-R | ArxivQA | Viz-cap | IconQA | CLEVR | Flickr30k | Avg   |
> | -| - | - | - | - | - | - | - |
> | SAME | 83.83   | 91.40   | 51.33   | 65.27  | 53.50 | 57.43  | 67.12 |
> | SAME w/ hard projection   | 81.12   | 91.88   | 48.43   | 52.62  | 39.75 | 56.53 | 61.72 |
> | SAME w/ constant subspace scaling | 81.75   | 91.25   | 48.19   | 53.12  | 40.88 | 55.80     | 61.83 |
>
> Our results confirm that SAME fundamentally differs from and significantly outperforms existing projection-based methods.
>
>
>
> [1] HiDe-LLaVA: Hierarchical decoupling for continual instruction tuning of multimodal large language model. ACL, 2025.

---

> > ### Author Rebuttal · Reviewer_QtnR · 2026-04-02
> >
> > Thank you for the response, which addressed some of our concerns. However, several issues remain:
> >
> > Data contamination: COIN was originally proposed for continual instruction tuning using LLaVA-1.5-base, whereas this work employs LLaVA-1.5-instruct. Since the instruct version has already undergone instruction tuning, data contamination risk still exists. Given that all experiments rely on COIN, this raises concerns about the credibility of the reported results.
> >
> > Novelty: We still find the technical contribution limited, primarily combining existing techniques without sufficiently novel ideas.
> >
> > Generalization: We suggest including experiments on OOD datasets (e.g., MMMU) to evaluate performance degradation after continual learning and compare with baselines.

---

> > > ### Author Response · Authors · 2026-04-03
> > >
> > > Thank you for the follow-up. We address the three remaining concerns below.
> > >
> > > ### 1) Data Contamination
> > >
> > > **SAME** follow the exact same training and evaluation procedure as **CoIN**. Specifically, we use the same model loading and initialization process, which follows the **CoIN protocol** (https://github.com/zackschen/CoIN/tree/CoIN?tab=readme-ov-file). This ensures that the comparison between **SAME** and the baselines is fair and consistent.
> > >
> > > Since **SAME** and all baselines are trained and evaluated using the same procedure and checkpoints, any pre-exposure to CoIN data affects **all methods equally**, ensuring that the results are meaningful. Moreover, **SAME improves CoIN average accuracy by +12.61%** over zero-shot, demonstrating that the gains are not simply due to pre-trained knowledge recall.
> > >
> > > To further address the contamination concern, we also evaluated **SAME on UCIT**, a benchmark that **strictly avoids data leakage**. **UCIT** tasks were never exposed to the same pre-training process, ensuring that the reported gains are genuinely due to the architecture itself. **SAME outperforms the baselines** on UCIT with the **same hyperparameters** as in CoIN.
> > >
> > > ### 2) Novelty
> > >
> > > We respectfully disagree with the characterization that SAME merely combines existing techniques without sufficient novelty.
> > >
> > > The novelty of SAME lies in how we **enforce stability in deep MLLMs**. Existing methods often rely on **hard null-space projection** or **constant isotropic scaling**. While effective in some contexts, these approaches face significant limitations in large-scale MLLMs. **Hard projection** restricts the model's flexibility by zeroing out certain directions, while **constant scaling** requires impractical tuning of hyperparameters, especially for models with many MoE layers.
> > >
> > > In contrast, **SAME** introduces **Spectral-aware Routing** for routers and **Curvature-aware Scaling** for experts, allowing fine-grained control over which directions are adapted, while **Adaptive Expert Activation** reduces interference and training cost.
> > >
> > > To address the reviewer’s concern, we replaced our proposed components with the methods from the two papers mentioned by the reviewer. These comparisons were performed on **UCIT**. The results show that **SAME outperforms these baselines by +5.40%** and **+5.29%** in average accuracy, respectively, demonstrating that **SAME** is both **mechanically** and **empirically** distinct from existing methods.
> > >
> > > **We respectfully suggest the reviewer to refer to our response to Q1 for the detailed results on UCIT.**
> > >
> > > ### 3) Generalization / OOD Evaluation
> > >
> > > We appreciate the suggestion to test OOD generalization. We looked into **MMMU**, but the official release is primarily an **evaluation benchmark**: the public release provides **150 development samples, 900 validation samples, and 10,500 test samples with withheld test answers**, across **six broad disciplines**, rather than a standard supervised training split for MCIT.
> > >
> > > To probe generalization further, we first present the results from evaluating **SAME using the UCIT-based checkpoint** on the **test sets of the six MMMU tasks**. The models, trained continually on **UCIT**, were directly evaluated on multiple tasks from the **MMMU** dataset, with the following performance:
> > >
> > > | Method    | Art & Design | Business | Science | Health & Med. | Humanities | Tech & Eng. | Avg.  |
> > > | --------- | ------------ | -------- | ------- | ------------- | ---------- | ----------- | ----- |
> > > | Zero-Shot | 43.85        | 25.35    | 23.74   | 32.93         | 48.47      | 28.16       | 33.75 |
> > > | MoELoRA   | 44.37        | 26.39    | 24.98   | 35.28         | 47.69      | 30.32       | 34.83 |
> > > | SAME      | 47.98        | 26.51    | 25.69   | 36.51         | 47.46      | 33.96       | 36.35 |
> > >
> > > In summary, **SAME** demonstrates strong performance after MCIT on **UCIT**, outperforming the baselines on the **MMMU** test tasks, highlighting its robust generalization capabilities.
> > >
> > > ---
> > >
> > > Additionally, we constructed a MCIT task using the six tasks from **MMMU**, which allows us to validate the model's MCIT ability directly on this dataset. We present the standard MCIT results using **MMMU's dev+valid sets** as the training data, and the test set as the evaluation benchmark:
> > >
> > > | Method    | Art & Design | Business | Science | Health & Med. | Humanities | Tech & Eng. | Avg.  |
> > > | --------- | ------------ | -------- | ------- | ------------- | ---------- | ----------- | ----- |
> > > | Zero-Shot | 43.85        | 25.35    | 23.74   | 32.93         | 48.47      | 28.16       | 33.75 |
> > > | MoELoRA   | 46.63        | 26.56    | 25.41   | 31.43         | 49.36      | 36.10       | 35.91 |
> > > | SAME      | 54.14        | 30.43    | 27.14   | 35.57         | 54.14      | 40.37       | 40.26 |
> > >
> > > Even in this much harder expert-level reasoning setting, **SAME** remains the strongest method.

---

### Official Review · Reviewer_P5xF · 2026-03-12

**Soundness:** 3
**Presentation:** 2
**Significance:** 3
**Originality:** 3
**Overall Recommendation:** 5
**Confidence:** 3

**Summary:**

This paper identifies two sources of forgetting in MoE-based multimodal continual instruction tuning — router drift and expert drift — and proposes SAME to address both. Experiments on CoIN show it outperforms existing methods.

**Compliance With Llm Reviewing Policy:**

Affirmed.

**Final Justification:**

The paper is good in its clear diagnostic experiment, well-grounded method design, and insightful analysis (especially the formatting-induced forgetting). The ablations are systematic and show each component is useful, and the approach also brings practical efficiency benefits.

**Key Questions For Authors:**

1. How does SAME perform under different task orderings? Even two or three random permutations would be informative.
2. What is the inference-time overhead? The paper only discusses training efficiency.
3. Are there plans to release the code?

**Limitations:**

yes

**Strengths And Weaknesses:**

Strengths

1. The diagnostic experiment in Figure 1 is well done. Freezing experts and retraining only the router to measure how much performance can be recovered is a neat way to show that router drift and expert drift are independent problems. This gives a clear motivation for the rest of the paper.

2. The technical design is formally grounded. I appreciate that the curvature-aware scaling is connected to Riemannian gradient descent rather than being presented as a pure heuristic, and the trick of coupling regularization strength to the learning rate schedule is nice since it avoids extra hyperparameters.

3. The formatting-induced forgetting analysis is an unexpected finding. The fact that so many ScienceQA errors come from letter-casing drift after learning TextVQA is something I haven't seen discussed elsewhere, and the "drop-rebound" pattern is informative.

4. The ablation in Table 2 is systematic and shows that each component contributes. Training speedups and memory savings from the expert freezing mechanism are a practical bonus.

Weaknesses

1. My main concern is that everything is evaluated on a single benchmark (CoIN), a single backbone, and a single fixed task ordering. Continual learning results can be quite sensitive to task ordering, so I would like to see at least a few random permutations. Without this, it is hard to know how much of the gain is robust vs. configuration-specific.

2. The paper claims "negligible memory overhead" for the covariance tracking, but never quantifies it. Given that spectral components and covariance matrices are stored per layer across all FFN layers, I suspect the overhead is not negligible for a model of this scale. Some numbers would help.

3. Several design choices feel underexplored. Why a sliding-window average for the scaling? Why subtraction-based freezing scores with min-max normalization rather than, say, a ratio? I would have liked to see at least one ablation comparing these choices against simpler alternatives. Also, the specific hyperparameter values (δ, μ, τ_score) are not reported anywhere that I could find, which makes it hard to judge how much tuning was needed.

4. Minor: the notation is heavy. Three components each introduce their own symbols and update rules, and it takes effort to keep track. A notation table would help. Also, Figure 2 tries to show everything at once and ends up quite cluttered — splitting it into separate sub-figures might be clearer.

---

> ### Author Rebuttal · Authors · 2026-03-31
>
> ## W1 & Q1: Robustness to task ordering & benchmark
>
> Thank your suggestion.
>
> Apart from CoIN, we also evaluate SAME on UCIT [1], a more challenging benchmark with stronger task heterogeneity. Using the same hyperparameters as on CoIN, SAME consistently outperforms the baselines:
>
> | Method| Avg |
> | -| -|
> | MoELoRA| 52.06|
> | HiDe-LLaVA | 64.19|
> | SAME| 67.12|
>
> We also test two additional task orderings on CoIN.
>
> Order 2:
>
> | Method | SciQA| Image| GQA | OCRVQA | REC  | VQAv2 | VizWiz | TextVQA | Avg  |
> | -| -| -| -| -| -| -| -| -| -|
> | MoELoRA  | 0  | 36.1  | 55.8 | 55.9 | 63.1 | 62.8  | 45.5   | 58.5  | 47.2 |
> | SAME | 78.1  | 89.7  | 60.8 | 63.4| 60.1 | 66.1  | 57.2 | 61.1| 67.0|
>
> Order 3:
>
> | Method | OCRVQA | VQAv2 | REC| VizWiz| GQA| Image | TextVQA | SciQA | Avg  |
> |-|-|-|-|-|-|-|-|- |-|
> | MoELoRA | 29.9 | 62.2 | 59.1| 47.4| 56.3| 45.3| 56.9 | 77.7| 54.3 |
> | SAME| 62.8 | 65.7| 58.6 | 53.9 | 61.3 | 90.1| 62.4 | 80.1| 66.8 |
>
> MoELoRA fails severely under ordering changes. In particular, the result on ScienceQA in Order 2 reflects complete forgetting rather than ordinary accuracy fluctuation. Though ScienceQA requires answer with uppercase option letters, 72.5% of errors are format mismatches (e.g., answering "july" rather than "A" for option "A.july"). This indicates MoELoRA forgets instruction-following ability.
>
> ## W2: Quantifying memory overhead
>
> Thank you for pointing this out. Storing a full covariance matrix $C$ requires 32 MB. In practice, following the randomized low-rank matrix decomposition [2], we compute an approximate factorization of $C$ and retain only the top-64 principal directions and singular values. Specifically, after obtaining the approximate decomposition $C \approx U\Sigma U^\top$, we store only $U\in\mathbb{R}^{d\times 64}$ and $\Sigma\in\mathbb{R}^{64}$. This reduces the storage for each covariance matrix to ~0.5 MB (a 64× compression), with only a 0.68% accuracy drop. At the full-model level, the entire SAME state fits within a 139 MB checkpoint.
>
> ## W3: Design choices and hyperparameters
>
> Thank you for the question.
>
> (a) Why a sliding-window average for the scaling?
>
> This design is crucial for SAME’s stability-plasticity balance. For the parallel subspace, existing methods either discard updates directly or scale all directions with a single constant, limiting plasticity and requiring careful tuning.
>
> In contrast, our sliding-window mechanism enables adaptive anisotropic scaling. Instead of a uniform scalar, SAME scales each singular direction using historical curvature, suppressing unstable directions while allowing updates in stable ones.
>
> Furthermore, the sliding window also captures local spectral density. When singular values are clustered, the exact placement of the energy threshold within this cluster has minimal impact, as the update for such directions is scaled by a factor close to 1. This significantly reduces sensitivity to compared to global averaging, which ignores this local structure.
>
> This yields a better stability-plasticity balance on UCIT:
>
> | Method|Avg. acc. |
> | -| - |
> | SAME w/ global mean scaling|63.3 |
> | SAME w/ sliding-window scaling (ours) |67.1 |
>
> (b) Why subtraction-based freezing scores with min-max normalization?
>
> We choose subtraction-based scoring over ratio-based scoring ($\tilde{U}/\tilde{F}$) for numerical stability: ratios become unstable when $F \approx 0$ (common in early tasks), making threshold selection difficult. Min-max normalization maps utilization and importance to a unified [0,1] scale for direct comparison, and by focusing on extreme values, enables more precise freezing decisions. We also compared against fixed-ratio freezing on UCIT:
>
> | Method| Avg. acc. |
> | -| -|
> | Fixed-Ratio (50%) Freezing | 60.4|
> | SAME| 67.1|
>
> (c) hyperparameter
>
> Due to limited computational resources, we randomly sampled 20% of each training dataset in UCIT for hyperparameter search.
>
> |Hyperparameter | Search Range| Final Value | Performance|
> | - | - | - | -|
> | $\delta$  | {0.80, 0.85, 0.90, 0.95} |0.90 | 66.5, 66.4, 67.4, 66.9 |
> | $\mu$  | {0.09, 0.9, 9}  | 0.9 | 62.3, 67.4, 61.4 |
> | $\tau_{\text{score}}$ | {0.075, 0.1, 0.125} | 0.1         | 65.7, 67.4, 66.3 |
>
> ## W4: Presentation / notation
>
> In the camera-ready version, we will add a notation table, report the missing hyperparameters, and split **Fig. 2 in the main paper** into separate sub-figures.
>
> ## Q2: Inference-time
>
> By mitigating router drift, SAME enables stable top-k routing during inference, whereas MoELoRA must avoid top-k selection due to routing instability, resulting in faster inference for SAME.
>
> | Method | Speed(it/s) |
> | -| - |
> | MoELoRA |1.03|
> | SAME |1.26|
>
> ## Q3: Code
>
> Yes. We plan to release the code and configs upon acceptance.
>
> [1] HiDe-LLaVA: Hierarchical decoupling for continual instruction tuning of multimodal large language model. ACL, 2025.
>
> [2] Finding structure with randomness: Probabilistic algorithms for constructing approximate matrix decompositions. SIAM Review, 2011.

---

> > ### Author Rebuttal · Reviewer_P5xF · 2026-04-05
> >
> > Thank you for the rebuttal and clarifications. After consideration, my overall assessment remains unchanged, and I will keep my original score.

---

> > > ### Author Response · Authors · 2026-04-05
> > >
> > > Thank you very much for your favorable assessment of our work and for your thoughtful feedback. We are glad to hear that our additional experiments and clarifications have addressed your concerns. We sincerely appreciate the time and effort you devoted to reviewing our work.

---

### Official Review · Reviewer_uc6N · 2026-03-13

**Soundness:** 2
**Presentation:** 2
**Significance:** 2
**Originality:** 2
**Overall Recommendation:** 4
**Confidence:** 2

**Summary:**

This paper studies multimodal continual instruction tuning in a rehearsal-free setting with Moe. The authors identify two sources of forgetting: router drift (changes in expert assignment over time) and expert drift (degradation of expert functionality due to continual updates).

**Compliance With Llm Reviewing Policy:**

Affirmed.

**Final Justification:**

The rebuttal improves clarity, and I appreciate the additional analysis provided. My overall assessment remains unchanged.

**Key Questions For Authors:**

1. To what extent is router drift an independent cause of forgetting rather than a consequence of representation changes in backbone/LoRA layers? Have you isolated these effects experimentally?

2. How sensitive is performance to the subspace rank and energy threshold? Have you tested scenarios with stronger task/domain shift?

3. Does adaptive expert activation lead to increasing task-specific specialization of experts over time? Have you analyzed expert-task affinity or utilization entropy?

**Limitations:**

The paper discusses practical aspects but could further elaborate on scalability to larger models and longer task sequences, as well as potential stability–plasticity trade-offs.

**Strengths And Weaknesses:**

The paper clearly distinguishes router drift and expert drift. The proposed stabilization framework is conceptually coherent.

The causal role of router drift in forgetting is not fully established, and it may partly reflect upstream representation shift.

The spectral subspace assumption behind routing stabilization could be sensitive to distribution shift and to the choice of hyperparameters.

Task-level expert freezing may implicitly promote task-specific expert partitioning, which could reduce the cross-task sharing benefits typically associated with Moe.

---

> ### Author Rebuttal · Authors · 2026-03-31
>
> ## W1 & Q1: Router drift causality
>
> Thank you for raising this question about the causal role of router drift. To isolate the router's impact more rigorously, we conduct a post-hoc controlled analysis: we trained SAME with the backbone frozen, saving the Task-1 expert snapshot immediately after Task 1. Crucially, for all subsequent tasks, both the router and the experts continue to be trained. After each task $t$, we evaluate performance on the Task-1 test set using the router from checkpoint ($t$) paired with the Task-1 experts. In this setup, with hidden representations and Task-1 experts fixed, performance variations stem solely from router weight changes, thereby isolating the causal effect of router drift.
>
> | Metric on Task 1 | T2 router | T3 router | T4 router | T5 router | T6 router | T7 router | T8 router |
> | - | - | - | - | -| -| - | - |
> | Routing overlap  | 0.933     | 0.929     | 0.926     | 0.931     | 0.934     | 0.912     | 0.927     |
> | Acc. drop (%)    | 0.8       | 1.1       | 1.4       | 1.6       | 1.5       | 2.1       | 1.3       |
>
> Even with Task-1 representations fixed, subsequent routers still diverge and degrade performance. This confirms router drift is an independent source of forgetting, not merely a byproduct of representation shifts.
>
> ## W2 & Q2: Sensitivity to spectral subspace assumptions and stronger task/domain shift
>
> We appreciate the concern that the spectral subspace assumption may be sensitive to hyperparameters and distribution shift.
>
> (a) Hyperparameter sensitivity. In SAME, the effective retained rank is induced by the energy threshold $\delta$. We therefore vary $\delta \in \{0.80,0.85,0.90,0.95\}$ and observe stable results on CoIN:
>
> | Energy threshold ($\delta$) | 0.80  | 0.85  | 0.90  | 0.95  |
> | - | -| - | -| -|
> | Final avg. acc. (%)         | 66.05 | 66.31 | 66.58 | 66.12 |
>
> The variation is below 1%, suggesting that SAME is not brittle to moderate changes in the retained spectral mass. This is consistent with our design: the scaling depends on relative local spectral structure rather than a hard rank cutoff, which reduces sensitivity to the exact threshold.
>
> (b) Stronger shift. We also evaluate SAME on UCIT using the same hyperparameters as CoIN, without retuning. UCIT has substantially larger cross-task heterogeneity [1]. Even in this stronger-shift setting, SAME remains the best performance:
>
> | Method     | Image-R   | ArxivQA   | Viz-cap   | IconQA    | CLEVR     | Flickr30k | Avg ↑     |
> | - | -| - | -| -| - | - | - |
> | MoELoRA    | 49.87     | 77.63     | 43.65     | 46.40     | 36.47     | **58.34** | 52.06     |
> | HiDe-LLaVA | 80.50     | 89.83     | 48.78     | 62.90     | 47.97     | 55.15     | 64.19     |
> | **SAME**   | **83.83** | **91.40** | **51.33** | **65.27** | **53.50** | 57.43     | **67.12** |
>
> Thus, SAME remains effective under stronger task/domain shift, and does not rely on delicate per-benchmark tuning.
>
> ## W3 & Q3: Does adaptive expert activation over-specialize experts?
>
> To examine expert specialization versus cross-task sharing, we ablate adaptive expert activation on CoIN by recording Layer-15 expert utilization and normalized routing entropy per task. Higher entropy indicates uniform sharing whereas lower entropy suggests specialization, quantifying whether experts become siloed or remain collaborative over time.
>
> | Task  | w/ Freezing | w/o Freezing |
> | - | - | - |
> | Task1 | 0.905  | 0.934|
> | Task2 | 0.803  | 0.914|
> | Task3 | 0.857  | 0.925|
> | Task4 | 0.860  | 0.906|
> | Task5 | 0.792  | 0.905|
> | Task6 | 0.668  | 0.905|
> | Task7 | 0.639  | 0.941|
> | Task8 | 0.478  | 0.956|
>
> Enabling adaptive expert activation decreases routing entropy, with the reduction magnitude increasing for later tasks. This trend reflects our adaptive freezing mechanism: as training progresses, more experts accumulate important knowledge from earlier tasks and are temporarily frozen to protect that knowledge. Consequently, routing becomes more concentrated on the remaining active experts, leading to lower entropy.
>
> This structured specialization pattern aligns with our design: experts are frozen to preserve knowledge transfer while reducing interference. The progressive entropy drop reflects adaptive allocation rather than random siloing, as experts dynamically adjust their participation based on task relevance and historical importance.
>
> ## Limitations
>
> We agree and will strengthen this discussion in the final version. Specifically, we will elaborate more clearly on scalability to larger models and longer task sequences, as well as potential stability–plasticity trade-offs. Our current results suggest that SAME achieves a favorable balance on both CoIN and UCIT, but this point should be discussed more explicitly.
>
>
>
> [1] HiDe-LLaVA: Hierarchical decoupling for continual instruction tuning of multimodal large language model. ACL, 2025.

---

> > ### Author Rebuttal · Reviewer_uc6N · 2026-04-04
> >
> > Thank you for the rebuttal and clarifications. My overall assessment remains unchanged, so I maintain my original score.

---

> > > ### Author Response · Authors · 2026-04-04
> > >
> > > Thank you for your kind response. We are glad to hear that our clarifications and experiments addressed your concerns. We sincerely appreciate the time and effort you devoted to reviewing our work.

---

### Decision · Program_Chairs · 2026-04-30

**Decision:**

Accept (regular)

**Comment:**

The paper proposes SAME (Stabilized Mixture-of-Experts) to address catastrophic forgetting in Multimodal Continual Instruction Tuning (MCIT). SAME aims to address "router drift" (inconsistent expert selection over time) and "expert drift" (overwriting of shared experts). To this end, SAME proposes 1) spectral aware routing that limits router updates using a low-rank covariance subspace decomposition, 2) curvature aware scaling that preconditions expert updates using inverse historical input covariance, and 3)adaptive expert activation/freezing that freezes experts that are low utility for the current task but historically important. Experiments on the CoIN benchmark report improved the final average over the MCIT baselines.

Reviewers found that the diagnostic analysis that disentangles router drift and expert drift provides novel insights. The proposed techniques are also well motivated: spectral aware routing stabilizes routing behavior, curvature aware scaling seeks to reduce destructive expert updates, and adaptive activation/freezing reduces unnecessary interference and improves efficiency. The ablation study and formatting-induced forgetting analysis are convincing. The empirical results on CoIN are strong. The rerouting protocol used to separate router drift from expert drift is meaningful.

Reviewers also raised concerns regarding the causal role of router drift, the spectral subspace assumption, lack of comparison and discussion of other design choices, evaluation on a single dataset, sensitivity of top-k expert selection, lack of forgetting metrics, the risk of data contamination, the fairness of comparing methods using base models vs. instruction finetuned models, the generalization capability changes on OOD benchmarks, etc. The authors successfully addressed most of them through the rebuttal, as reflected by the "fully resolved" feedback from three reviewers. However, the fairness concern was not fully solved, and more clarification about the setups for different methods is needed. Moreover, the theoretical justification for spectral-aware routing is not fully rigorous. Therefore, I suggest a weak acceptance.